Technical Report

# Studying the genetics of participation using footprints left on the ascertained genotypes

Stefania Benonisdottir [1] ✉ & Augustine Kong [1,2] ✉

The trait of participating in a genetic study probably has a genetic component. Identifying this component is difficult as we cannot compare genetic information of participants with nonparticipants directly, the latter being unavailable. Here, we show that alleles that are more common in participants than nonparticipants would be further enriched in genetic segments shared by two related participants. Genome-wide analysis was performed by comparing allele frequencies in shared and not-shared genetic segments of first-degree relative pairs of the UK Biobank. In nonoverlapping samples, a polygenic score constructed from that analysis is significantly associated with educational attainment, body mass index and being invited to a dietary study. The estimated correlation between the genetic components underlying participation in UK Biobank and educational attainment is estimated to be 36.6%—substantial but far from total. Taking participation behaviour into account would improve the analyses of the study data, including those of health traits.

For all sample surveys, ascertainment bias, meaning that the sample is not representative of the target population, could lead to seriously misleading conclusions[1,2]. By its very nature, ascertainment bias usually cannot be evaluated based on the sample alone[3]. Typically, other variables (covariates) that have known distributions for both sample and population are needed for adjustments[1,3]. Such adjustments are inherently imperfect as the covariates are unlikely to fully capture the underlying bias[1,3]. For genetic studies, among participants of the primary study who have contributed DNA, further engagement in optional components of the study has been demonstrated to have associations with genotypes and phenotypes[4–7]. That, however, does not address the genotypic difference between the primary study participants and the target population. Thus, it is striking that one can investigate how the sampled genotypes are biased based on themselves alone. A recent study identified single nucleotide polymorphisms (SNPs) that had significant allele frequency differences between the sampled males and females, and proposed that those variants have differential participation effects for the sexes[8]. This approach, however, cannot identify variants that affect primary study participation of both sexes in a similar manner. Here, we show how to do so.

## Results

### Three allele comparisons

All individuals are genetically related to some degree. Furthermore, each individual has two copies of genetic segments on autosomal chromosomes, and some of these segments are identical by descent (IBD), that is inherited from a recent common ancestor, with genetic segments in a relative. Instead of comparing individuals, we compare genetic segments. The key idea is that an allele that has higher frequency in participants than nonparticipants would also have higher frequency in segments that are in two participants than in segments that are in only one. Following this observation, we present below three principles of genetic induced participation bias, and show how to use only the sampled genetic data to perform genome-wide association scans (GWAS) for study participation that capture only direct genetic effects[9,10], and are unaffected by population stratification[11].

**First principle of genetic induced ascertainment bias.** On average, between two ascertained individuals, genetic segments shared IBD, relative to segments that are not, are enriched with alleles that have positive direct effects on ascertainment probability. Figure 1a illustrates this

[1]Big Data Institute, Li Ka Shing Centre for Health Information Discovery, University of Oxford, Oxford, UK. [2]Leverhulme Centre for Demographic Science, University of Oxford, Oxford, UK. ✉e-mail: stefania.benonisdottir@gmail.com; augustine.kong@bdi.ox.ac.uk

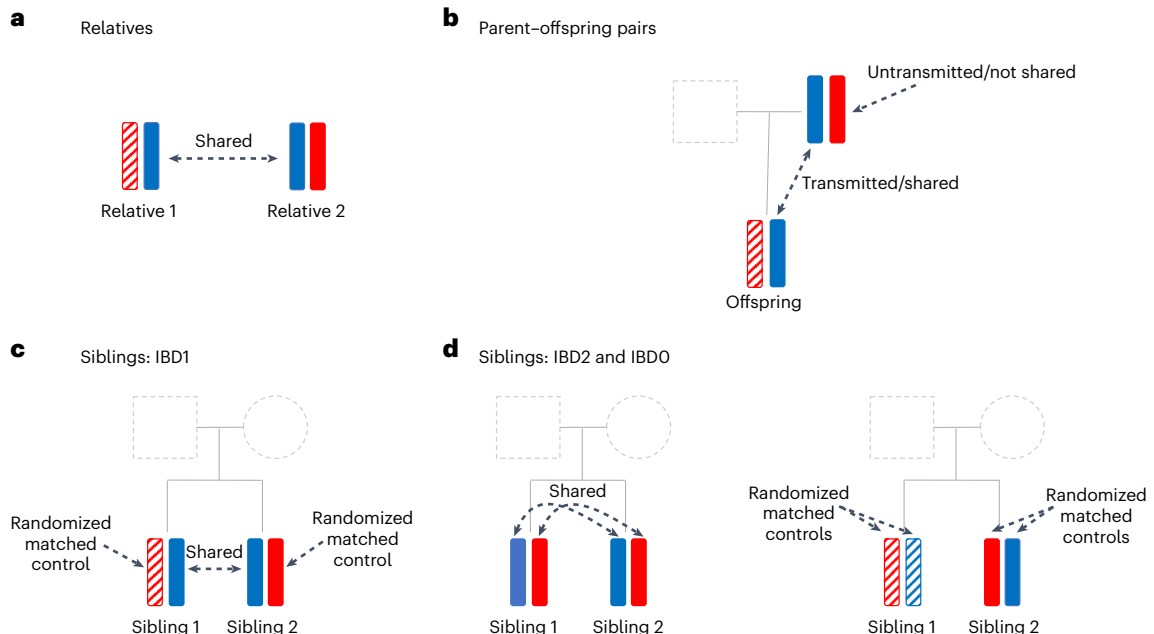

**Fig. 1 | The first principle of genetic induced participation bias: comparing shared and not-shared alleles. a**, General relative pairs sharing one allele IBD.
**b**, Parent–offspring pairs. **c**, Sib-pairs sharing one allele IBD (IBD1). **d**, Sib-pairs sharing two alleles or no allele IBD (IBD2 and IBD0). Different shading denotes segments that are distinct by descent.

general principle. With a large sample of individuals of the same ancestry, at a specific SNP locus, many pairs of individuals would share one long haplotype inherited IBD from a not-very-distant common ancestor. Each of such pairs has one distinct shared haplotype, and two distinct not-shared haplotypes. The SNP allele that promotes participation would tend to have a higher frequency in the shared than the not-shared haplotypes. The shared and not-shared alleles are considered as cases and controls, respectively, and matched as they are in the same individuals. Still, that does not remove potential confounding entirely as haplotypes driven to higher frequency through natural selection would also be shared by more individuals. Ascertainment bias is a form of selection and, to cleanly distinguish it from other forms of selection, requires more stringent matching of shared and not-shared haplotypes. That is achieved by using ascertained parent–offspring and sibling pairs (sib-pairs).

**Ascertained parent–offspring pairs.** For an ascertained parent–offspring pair (Fig. 1b) where the other parent is not ascertained, there are three distinct alleles by descent for a given SNP: the allele transmitted from parent to offspring ($T$), the parental allele not transmitted to the offspring (NT) and the allele inherited by the offspring from the other parent ($O$). The $T$ allele is shared, and the other two are not-shared. The NT and $T$ alleles are perfectly matched as both are in the ascertained parent. Mendelian inheritance dictates that each would have the same chance to be transmitted to the offspring to become the shared allele. The principle is similar to that underlying the transmission disequilibrium test[12]. The $O$ allele is not as perfectly matched and is currently not used. For the ascertained parent–offspring pairs, with alleles coded as 0/1, and $F_T$ and $F_{NT}$ denoting the frequency of 1s among the $T$ and NT alleles respectively, the difference

$$F_T - F_{NT} \qquad (1)$$

can be used to test for association between SNP and ascertainment (Methods). We call this the transmitted–nontransmitted comparison (TNTC). When both parents are ascertained, that can be treated as two parent–offspring pairs as the transmissions from the two parents are independent without a participation effect.

**Ascertained sib-pairs, parents not ascertained.** At a specific locus, assuming random Mendelian transmission, a sib-pair has probability 1/2 of inheriting the same allele IBD from the father, and the same independently from the mother. Consequentially, a sib-pair shares 2, 1 or 0 alleles IBD with probabilities 1/4, 1/2 and 1/4, respectively. With dense SNPs, the IBD state of a locus can usually be determined with high accuracy. For a specific SNP, based on the sib-pairs with known IBD states, two comparisons are made.

For each sib-pair that has IBD state 1 for a given SNP (Fig. 1c), there are one distinct shared allele ($S$) and two distinct not-shared alleles (NS$_1$ and NS$_2$). Let $F_{IBD1S}$ and $F_{IBD1NS}$ denote the frequency of allele 1 among the $S$ alleles and the NS (NS$_1$ and NS$_2$ combined) alleles, respectively. The difference

$$F_{IBD1S} - F_{IBD1NS} \qquad (2)$$

is the within-sib-pair comparison (WSPC). For any such pair, if the $S$ allele is paternally inherited, then the two NS alleles are maternally inherited, and vice versa. Despite that, conditional on the genotypes of the two parents, without ascertainment bias, the frequency difference has expectation 0. Assuming random transmissions, the shared allele is equally likely to be paternal or maternal (Extended Data Fig. 1). Thus, any systematic difference between fathers and mothers cancels in expectation.

For the sib-pairs sharing two alleles IBD at a locus, let $F_{IBD2}$ be the frequency of allele 1. Similarly, let $F_{IBD0}$ be the allele 1 frequency among sib-pairs that share no allele IBD. The difference (Fig. 1d)

$$F_{IBD2} - F_{IBD0} \qquad (3)$$

is the between-sib-pairs comparison (BSPC). This compares different sib-pairs and does not require splitting genotypes into shared and not-shared alleles. For a sib-pair at a particular locus, the chance to be in IBD state 0 or 2 is the same and, thus, without ascertainment bias, the frequency difference has expectation zero. In general, a sib-pair would be in IBD state 2 for some regions, and in IBD state 0 (or 1) in other regions.

Results from the three comparisons introduced can be combined, for testing, estimation or prediction purposes. However, having the

individual results separately is important because they could capture different effects depending on the nature of the participation bias. Most importantly, they are impacted differently by genotyping and data-processing errors.

## Analyses of UK Biobank data

The UK Biobank (UKBB) is a large-scale database with genetic and phenotypic information of individuals from across the UK[13]. Individuals aged between 40 and 69 years and living within a 25-mile radius of any of the 22 UKBB assessment centers were invited to participate[14]. Among the 9,238,453 invited, 5.5% (~500,000 individuals) participated and went through baseline assessments that took place from 2006 to 2010 (ref. 14). In addition to phenotypic details collected at the baseline visit, information continued to be added, including follow-up studies for large subsets of the cohort[14–16]. It is known that the UKBB sample is not fully representative of the UK population[13,14,17]. Participants were more likely to be female, less likely to smoke, older, taller, had lower body mass index (BMI)[14] and more educated[17].

We applied our methods to 4,427 parent–offspring pairs and 16,668 sib-pairs with White British descent in UKBB (Supplementary Note section 1; Methods). To start, association analysis was performed for 658,565 SNPs available in the UKBB phased haplotype data[13]. For each SNP, $t$-statistics for TNTC, WSPC and BSPC were computed by dividing each of the frequency difference, equations (1) to (3), by its standard error (s.e.) (Supplementary Note section 2; Methods). When results from all the SNPs were examined together, we observed a tendency for the major allele to be positively associated with participation. Investigations supported by extensive simulations revealed three causes to this bias: (a) IBD-calling errors, (b) phasing errors, (c) genotyping errors (Methods). (a) affects the processing of sib-pair data when the IBD sharing status (0, 1 or 2) of every SNP is 'called' and impacts WSPC and BSPC. The problem with (a) was mostly eliminated when we replaced KING[18], a 'general' IBD-calling program, by snipar[9], specifically designed for sib-pairs (Extended Data Fig. 2), and trimmed away 250 SNPs from each end of an IBD segment (contiguous SNPs with the same IBD status called; Extended Data Fig. 3; Methods). (b) and (c) affect TNTC and WSPC, which require splitting genotypes into shared and not-shared alleles; (b) enters when the relatives are both heterozygous and phasing with neighboring SNPs is required to determine the shared allele[9] (Extended Data Fig. 4). Through theoretical calculations and simulations, we found that under most scenarios, (b) and (c) would contribute to the observed major-allele bias (Supplementary Note section 3; Methods). For variants with minor allele frequency (MAF) > 0.10, we estimated that around 50% of the observed major-allele bias can be explained by phasing errors, while genotyping errors play a greater role for low frequency variants (Extended Data Figs. 5 and 6; Methods). As (b) and (c) do not impact BSPC and with (a) addressed, results based on BSPC can be used in a straightforward manner. When LD score regression (LDSC)[19] is applied to the $\chi^2$ statistics computed from the BSPC GWAS, the fitted intercept is nearly exactly 1 (0.9998), indicating that the $\chi^2$ statistics are neither inflated because of data artefacts, nor are they affected by issues such as population stratification. For TNTC and WSPC, the major-allele bias and related problems affect different analyses differently, as illustrated by analyses below and information in Methods and Supplementary Note section 3. For most SNPs individually, the bias induced by (b) and (c) is small and apparent only when analyzed as a group. However, in the major histocompatibility complex (MHC), a difficult region with extended linkage disequilibrium (LD), 17 SNPs have $P < 5 \times 10^{-8}$ with WSPC. These are clearly data artefacts as none is even nominally significant with BSPC (all $P$ values presented are two-sided). We discarded SNPs from the MHC and other extended LD regions[20] and applied other quality control filtering (Supplementary Note section 4; Methods), leaving 500,632 remaining SNPs with MAF > 1% in our participation genome scans. To handle the data-error induced biases

separately for TNTC and WSPC, a two-step allele frequency adjustment was applied, which eliminated the correlation with allele frequency and shrank the $t$-statistics towards zero (Extended Data Fig. 7; Methods). Figure 2 summarizes the steps underlying the procedure used to test SNPs for association with participation.

GWAS based on BSPC does not give any genome-wide significant SNP, which is unsurprising given that the power is not high. Combining the association results from all three comparisons, one SNP, rs113001936 on chromosome 16 ($P = 3.4 \times 10^{-9}$), passed the genome-wide significant threshold. However, since the positively associated allele has a frequency of 0.988, and the significance is driven mainly by WSPC ($P = 0.048$ for BSPC), we consider this SNP as only suggestive. To validate and to provide a proper understanding of the results, we constructed three separate participation polygenic scores (pPGSs) whose SNP-weights are based on these three sets of $t$-statistics (Methods). Values of each of the three pPGSs, standardized to have variance 1, were computed for 272,409 White British individuals without relatives in UKBB of third degree or closer, referred to as the 'unrelateds'. Associations between each of the three pPGSs and various quantitative traits were examined using the unrelateds (Supplementary Note section 5; Methods). Table 1 shows some of the strongest associations plus a few nonsignificant ones for reference. The BSPC pPGS and the WSPC pPGS are supposed to estimate essentially the same effects with comparable power (see simulation results below). Thus, it is comforting that their associations with the various traits are generally compatible. With multiple-comparison adjustment, the difference between the BSPC and the WSPC pPGS associations is not significant for any of the traits. Considering that the TNTC pPGS is based on a smaller sample size, its association results are also in general agreement. This shows that the data errors impacting TNTC and WSPC have only a small effect on polygenic score prediction. Indeed, even without adjustments, the TNTC and WSPC pPGSs give associations (Supplementary Table 1) similar to those in Table 1.

We constructed a combined pPGS using the results from all three comparisons (Methods). Its strongest association is with educational attainment (EA) where the effect (in standard deviation (s.d.) units) is 0.0309 with $P = 3.9 \times 10^{-53}$. The effect, 0.0300, is nearly as strong for age-at-first-birth (AFB) of women ($P = 8.6 \times 10^{-21}$). The next strongest association, notably negative, is with BMI ($P = 4.0 \times 10^{-22}$). These results, consistent with the known differences in EA and BMI between the UKBB sample and population[14,17], validated that our GWAS performed without phenotype data can nonetheless capture genetic associations with participation.

Some UKBB participants were invited to answer a dietary study questionnaire in 2011–2012 (ref. 16), and some were invited to a physical activity study in 2013–2015 that required wearing an accelerometer for a week[15]. Not everybody was invited (criteria included having a valid email address) and only a subset of those invited actually participated. We call participation in these follow-up studies 'secondary participation'. For the dietary study, the estimated effect of the combined pPGS on being invited, in $\log_e$ odds-ratio ($\log(OR)$), is 0.0342 ($P = 4.8 \times 10^{-16}$; Table 1). For those invited, the estimated effect on actual participation in $\log(OR)$ is 0.0272 ($P = 2.5 \times 10^{-7}$). For the physical activity study, the corresponding estimates are 0.0277 ($P = 1.1 \times 10^{-12}$) and 0.0300 ($P = 6.4 \times 10^{-8}$).

Associations with the combined pPGS were further examined adjusting for EA (Table 1). For traits/variables with $P < 1 \times 10^{-3}$ originally, the adjusted effects shrink but all remain significant. Relatively, the effect on participating in the physical activity study when invited shrinks the least, by 18%, from 0.0300 to 0.0246, while the effect on dietary study invitation shrinks by 38%, from 0.0342 to 0.0213. This difference in shrinkage is partly because the estimated effect of EA on dietary study invitation is 0.4661, much larger than its effect on physical activity study participation, 0.1826. From the EA effects alone, the ascertainment bias would seem to be much stronger with dietary study

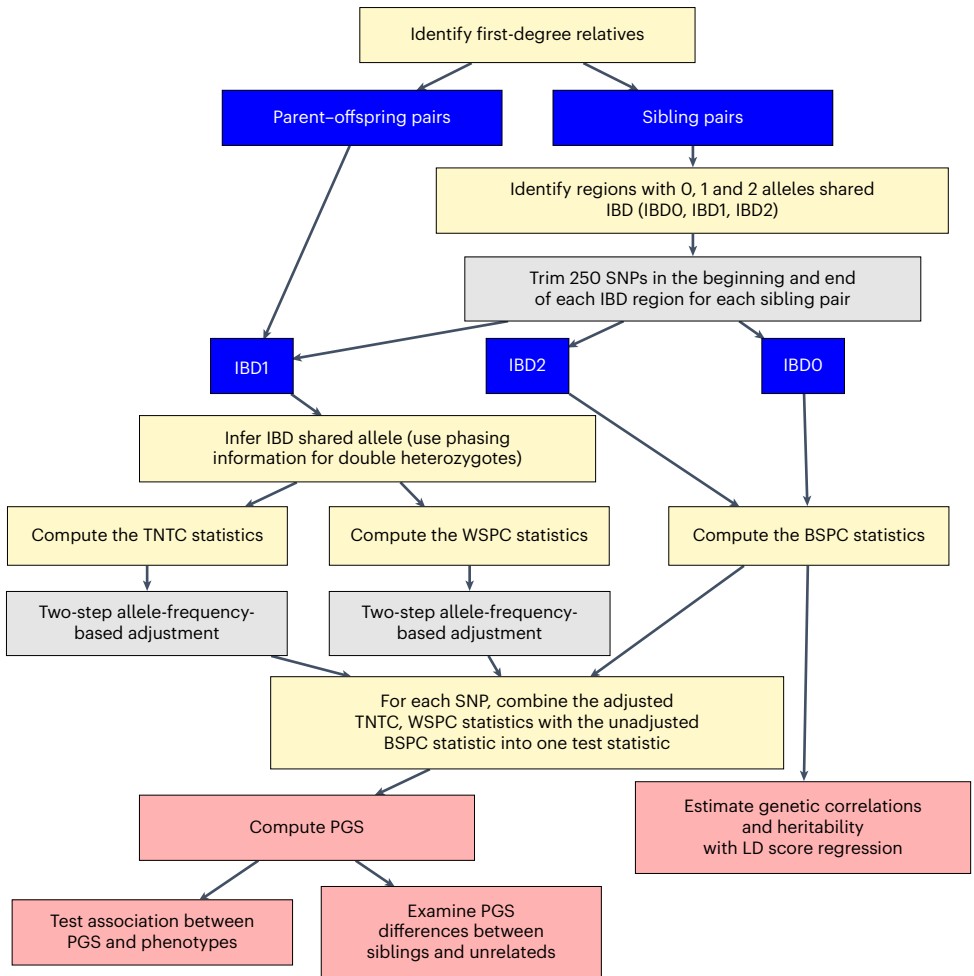

**Fig. 2 | Flowchart summarizing the procedure implemented to test genetic variants for association with primary participation using genotypes of participants only.** Core steps are shown as yellow rectangles. The procedure involves dividing the data into different data groups, shown here as blue rectangles. Gray rectangles show additional quality control steps, implemented to reduce or adjust for genotyping and data-processing errors. Pink rectangles show within-study validation steps.

invitation. By contrast, the genetic component of the ascertainment bias for physical study participation is stronger than that for dietary study invitation after EA adjustment. Thus, while the participation genetic component is associated with EA, its effect on other traits is not manifested mainly through EA. Importantly, phenotypes known to correlate with participation do not fully capture the nature and magnitude of the ascertainment bias. When males and females are analyzed separately for the traits/variables in Table 1 (Fig. 3), no significant difference is found for the pPGS effects with multiple-comparison adjustment. Furthermore, while UKBB participation rates differ by sex and age[14], the combined pPGS is associated with neither ($P > 0.05$), implying that the effect of the combined pPGS is additive to the effects of sex and age on participation, with no detectable statistical interactions. This, however, does not rule out the existence of variants with strong differential effects for the sexes as the combined pPGS is based on GWASs that are not sex specific and thus not designed to capture such variants.

Advancing from hypothesis testing to parameter estimation, we note that the UKBB did not recruit families, and participants were all adults providing their own consent[13]. Under these conditions, for alleles associated with participation, relative frequencies in different groups of individuals and genetic segments depend on many factors, most important of which are overall participation rate, denoted here by $\alpha$, and the participation rate of close relatives of participants (Fig. 4 and

Extended Data Fig. 8). For sib-pairs, the participation rate of an individual given that its sibling participates divided by $\alpha$ is the sibling recurrence participation rate ratio, denoted by $\lambda_S$. Simulating from a simplified scenario where the population consists of a set of sib-pairs, we examined the relationships between allele frequencies in various groups of individuals and segments (Fig. 4 and Extended Data Fig. 8; Methods). We assumed a liability-threshold model where a person participates if the liability score is above a certain threshold (Methods). Given $\alpha$, the correlation of siblings' liability scores, induced by both genetic and nongenetic factors, determines $\lambda_S$. Results based on assuming $\alpha = 5.5\%$, the participation rate of UKBB, are presented here. For an SNP, let $f_{pop}$ and $F_{samp}$ denote respectively the frequency of allele 1 in the population and in the participants. Allele 1 is assumed to have a positive participation effect. Results highlighted here are frequency differences relative to ($F_{samp} - f_{pop}$). The latter has the following relationship with the frequency difference between participants and nonparticipants:

$$(F_{samp} - f_{pop}) = (1 - \alpha)(F_{samp} - F_{nonparticipants}). \qquad (4)$$

In Fig. 4, the simulated averages of the ratios

$$\frac{F_{IBD1S} - F_{IBD1NS}}{F_{samp} - f_{pop}} \qquad (5)$$

**Table 1 | Primary pPGSs association with phenotypes**

| Phenotypes | TNTC pPGS | | WSPC pPGS | | BSPC pPGS | | Combined weights pPGS | | Combined weights pPGS adjusted for EA | | Sample size |
|---|---|---|---|---|---|---|---|---|---|---|---|
| Quantitative | Effect | Pvalue | Effect | Pvalue | Effect | Pvalue | Effect | Pvalue | Effect | Pvalue | N |
| EA | 0.0094 | $3.5\times10^{-6}$ | 0.0231 | $3.0\times10^{-30}$ | 0.0198 | $1.1\times10^{-22}$ | 0.0309 | $3.9\times10^{-53}$ | - | - | 260,950 |
| AFB (women) | 0.0084 | $8.9\times10^{-3}$ | 0.0196 | $1.1\times10^{-9}$ | 0.0223 | $3.4\times10^{-12}$ | 0.0300 | $8.6\times10^{-21}$ | 0.0197 | $1.1\times10^{-10}$ | 98,653 |
| BMI | −0.0112 | $9.9\times10^{-9}$ | −0.0077 | $8.1\times10^{-5}$ | −0.0147 | $4.5\times10^{-14}$ | −0.0189 | $4.0\times10^{-22}$ | −0.0148 | $6.5\times10^{-14}$ | 271,535 |
| HDL cholesterol | 0.0038 | $7.5\times10^{-2}$ | 0.0065 | $2.2\times10^{-3}$ | 0.0095 | $7.8\times10^{-6}$ | 0.0118 | $3.0\times10^{-8}$ | 0.0085 | $7.5\times10^{-5}$ | 237,785 |
| Height | 0.0035 | $9.1\times10^{-2}$ | 0.0089 | $1.5\times10^{-5}$ | 0.0028 | 0.17 | 0.0087 | $2.1\times10^{-5}$ | 0.0043 | $3.6\times10^{-2}$ | 271,820 |
| Glycated hemoglobin | −0.0072 | $3.7\times10^{-4}$ | 0.0009 | 0.66 | −0.0057 | $5.2\times10^{-3}$ | −0.0060 | $3.0\times10^{-3}$ | −0.0039 | $5.9\times10^{-2}$ | 259,594 |
| Number of siblings | −0.0042 | $3.9\times10^{-2}$ | −0.0044 | $3.1\times10^{-2}$ | −0.0008 | 0.69 | −0.0050 | $1.4\times10^{-2}$ | −0.0003 | 0.89 | 268,191 |
| Number of children | −0.0013 | 0.51 | −0.0037 | $5.7\times10^{-2}$ | −0.0029 | 0.13 | −0.0047 | $1.5\times10^{-2}$ | −0.0030 | 0.13 | 271,317 |
| Glucose | −0.0006 | 0.79 | −0.0041 | $4.6\times10^{-2}$ | −0.0020 | 0.33 | −0.0041 | $4.9\times10^{-2}$ | −0.0028 | 0.18 | 237,629 |
| Vitamin D | −0.0006 | 0.75 | −0.0019 | 0.35 | −0.0010 | 0.63 | −0.0020 | 0.32 | −0.0005 | 0.82 | 249,079 |
| Sex hormone binding globulin | 0.0074 | $4.1\times10^{-4}$ | 0.0021 | 0.31 | −0.0039 | $6.2\times10^{-2}$ | 0.0019 | 0.37 | 0.0020 | 0.35 | 235,960 |
| Grip strength | −0.0024 | 0.21 | 0.0010 | 0.62 | −0.0009 | 0.63 | −0.0010 | 0.60 | −0.0023 | 0.24 | 270,525 |
| Lipoprotein A | −0.0017 | 0.41 | −0.0026 | 0.20 | 0.0022 | 0.27 | −0.0009 | 0.66 | −0.0005 | 0.82 | 251,698 |
| Testosterone | 0.0031 | 0.11 | 0.0023 | 0.25 | −0.0031 | 0.12 | 0.0008 | 0.70 | 0.0009 | 0.65 | 257,570 |
| **Binary** | log(OR) | Pvalue | log(OR) | Pvalue | log(OR) | Pvalue | log(OR) | Pvalue | log(OR) | Pvalue | $N_{case}/N_{control}$ |
| Dietary study invitation | 0.0091 | $3.1\times10^{-2}$ | 0.0244 | $7.2\times10^{-9}$ | 0.0237 | $1.9\times10^{-8}$ | 0.0342 | $4.8\times10^{-16}$ | 0.0213 | $1.3\times10^{-6}$ | 166,993/105,416 |
| Dietary study participation | 0.0150 | $4.3\times10^{-3}$ | 0.0123 | $2.0\times10^{-2}$ | 0.0207 | $8.4\times10^{-5}$ | 0.0272 | $2.5\times10^{-7}$ | 0.0199 | $2.4\times10^{-4}$ | 54,124/112,869 |
| Physical activity study invitation | 0.0137 | $4.2\times10^{-4}$ | 0.0163 | $3.0\times10^{-5}$ | 0.0184 | $2.2\times10^{-6}$ | 0.0277 | $1.1\times10^{-12}$ | 0.0182 | $6.5\times10^{-6}$ | 132,633/139,776 |
| Physical activity study participation | 0.0170 | $2.2\times10^{-3}$ | 0.0203 | $2.4\times10^{-4}$ | 0.0159 | $4.2\times10^{-3}$ | 0.0300 | $6.4\times10^{-8}$ | 0.0246 | $1.6\times10^{-5}$ | 59,455/73,178 |

Regression was used for analyzing quantitative traits and logistic regression was used for binary traits. Displayed are fitted coefficients (Effect/log(OR)) and Pvalues from regressing phenotypes on the pPGS in a subset of White British unrelateds, taking sex, year of birth, age at recruitment, genotyping array and 40 principal components (PCs) into account (Methods). Grip strength was additionally adjusted for height. The pPGS and the quantitative phenotypes were transformed to have a variance of 1 and thus effect is in s.d. units. When computing Pvalues, to account for population stratification, the standard errors of the test statistics (t-tests for quantitative phenotypes and z-tests for binary phenotypes) are adjusted using the LD score regression intercept (Methods). Pvalues are two-sided without multiple-comparison correction. Association results for four different pPGSs are shown with weights derived from (1) TNTC with a two-step allele-frequency-based adjustment, (2) WSPC with a two-step allele-frequency-based adjustment, (3) BSPC and (4) a linear combination of TNTC, WSPC and BSPC results (Combined weights). Also shown are association results for the combined pPGS adjusted for educational attainment (EA).

and

$$\frac{F_{IBD2} - F_{IBD0}}{F_{samp} - f_{pop}} \quad (6)$$

are nearly identical, indicating that WSPC and BSPC are capturing real effects in a similar manner. Both ratios are close to 1 when $\lambda_S$ is close to 1, and decrease gradually as $\lambda_S$ increases. For $\lambda_S = 2$, the reported UKBB sib-pair enrichment, the ratios are around 0.86. Results here are simulated from an SNP with $f_{pop} = 0.5$ and with effect accounting for 0.1% of the variance of the liability score. Ratios for other parameter values are similar if the individual SNP effect is small. Results in Fig. 4 are for the genotype having an additive effect. Assuming allele 1 has a dominant effect, $(F_{IBD2} - F_{IBD0})$, is about 10% smaller than $(F_{IBD1s} - F_{IBD1NS})$. For a recessive model, $(F_{IBD1s} - F_{IBD1NS})$ is about 8% smaller than $(F_{IBD2} - F_{IBD0})$. Based on the same set of sib-pairs, BSPC and WSPC are directly comparable. TNTC is based on a separate set of participant pairs and thus relative effects of SNPs can differ. Reasons include the parents being older and the asymmetric relationship between parent and offspring (Supplementary Note section 6).

Alleles in the unrelateds are not IBD shared through a recent common ancestor with any other participant, while participants with close relatives participating have both shared and not-shared alleles. This leads to the second principle.

**Second principle of genetic induced ascertainment bias.** The second principle is that alleles that promote participation would have different frequencies in participants with participating close relatives and in those without. In most cases, the frequencies in the former would be higher. From the same simulations, Fig. 4 displays the simulated average of the ratio

$$\frac{F_{SIBS} - F_{SING}}{F_{samp} - f_{pop}} \quad (7)$$

where $F_{SIBS}$ is the allele frequency in the participating sibling pairs, and $F_{SING}$ (SING denotes 'singletons') is the allele frequency in the participating individuals whose sibling does not participate. Examining this empirically without overfitting, we randomly partitioned the UKBB sibling pairs into two halves. The pPGS with weights derived from the GWASs using data of the first half ($pPGS_1$) has average value for the second half of the sibling pairs that is 0.044 s.d. higher than that of the unrelateds ($P = 5.0 \times 10^{-6}$). In reverse, the first half of the sibling pairs have average $pPGS_2$ value, weights derived from the second half, that is 0.026 s.d. higher than the unrelateds ($P = 8.5 \times 10^{-3}$).

**Genetic correlation and heritability.** We applied LD score regression using LDSC[21] to estimate the correlations between the genetic component underlying primary participation and the genetic components of

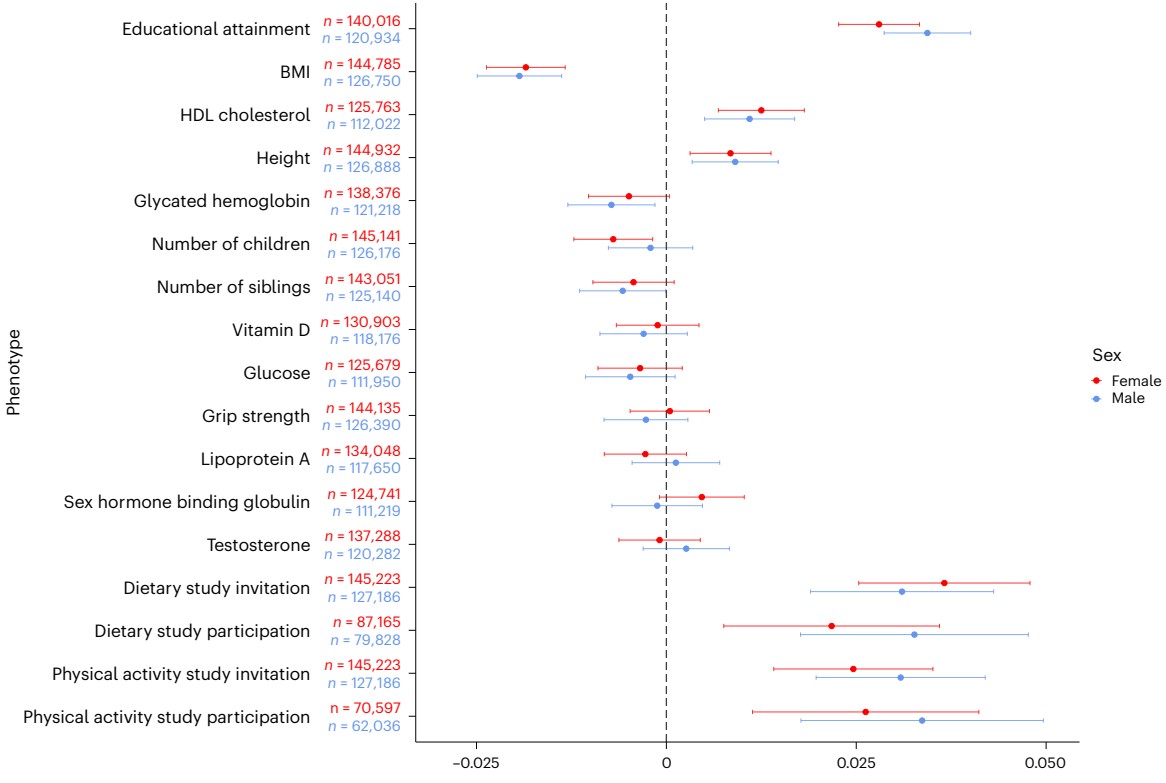

**Fig. 3 | Sex-specific analysis of pPGS associations.** Centers of error bars correspond to effect estimates in Table 1 for the pPGS based on combined weights; here, the association analyses were performed for male (blue error bars) and female (red error bars) unrelateds separately. Error bars correspond to 95% CI (estimate ±1.96 s.e.). Quantitative phenotypes were standardized to have a variance of 1 in males and females separately (Supplementary Note section 5). For secondary participation traits, the sample size shown equals number of cases plus controls.

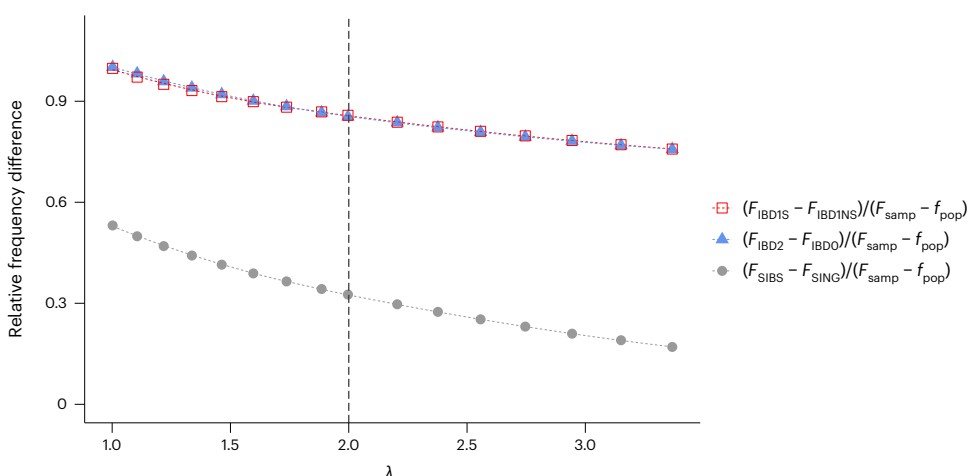

**Fig. 4 | Relative frequency differences as functions of sib-pairs enrichment in sample.** For $\alpha = 0.055$, the participation rate of UKBB, three frequency difference ratios of an SNP (indicated in the figure) with participation effects are displayed as functions of the sibling recurrence participation rate ratio, $\lambda_S$. The results are from simulations under a liability-threshold model where the participation of an individual is determined by its liability score and the participation rate $\alpha$. Given $\alpha$, $\lambda_S$ is a function of the correlation between the liability scores of two siblings (Methods). In particular, for $\alpha = 0.055$, a correlation of 0.193 between the siblings' liability scores leads to $\lambda_S = 2$, the reported enrichment of sib-pairs in the UKBB data. We simulated 500 replications from a population of $5 \times 10^7$ sib-pairs. Allele 1 of the SNP is assumed to have a population frequency of 0.5 and the effect of the SNP is assumed to account for 0.1% of the variance of the liability score. The simulated averages of the two ratios, $(F_{IBD1S} - F_{IBD1NS})/(F_{samp} - f_{pop})$ and $(F_{IBD2} - F_{IBD0}) / (F_{samp} - f_{pop})$, shown with red hollow squares and blue solid triangles respectively, are virtually indistinguishable from each other; they are roughly 1 when $\lambda_S$ is close to 1 and decrease gradually as $\lambda_S$ increases, but are always positive. The simulated average of the third ratio $(F_{SIBS} - F_{SING})/(F_{samp} - f_{pop})$, where $F_{SIBS}$ is the allele frequency in the participating sibling pairs and $F_{SING}$ is the allele frequency in the participating individuals whose sibling does not participate, is shown with gray solid circles. For $\lambda_S = 2$, the first two ratios are around 0.86 and the third ratio is around 0.32.

**Table 2 | Estimates of genetic correlations between primary participation and other traits**

| Phenotype | $\rho$ | (s.e.) |
|---|---|---|
| EA | 0.366 | (0.096) |
| AFB (women) | 0.455 | (0.134) |
| BMI | −0.219 | (0.073) |
| Dietary study invitation | 0.418 | (0.125) |
| Dietary study participation | 0.388 | (0.148) |
| Physical activity study invitation | 0.426 | (0.147) |
| Physical activity study participation | 0.366 | (0.160) |

Genetic correlations with primary participation ($\rho$) estimated with LD score regression using the program LDSC (Methods). Results here are for traits in Table 1 where the 95% CIs of genetic correlation do not include zero. The analyses were performed with the European Ancestry LD scores computed by the Pan-UKB team[23] and were restricted to the 500,632 SNPs used to compute the pPGSs with the exact number of SNPs ranging from 496,517 to 497,378. For primary participation, the GWAS results from the BSPC were used. For the other traits, summary statistics were obtained by performing a GWAS based on the 272,409 White British unrelateds with available phenotype values (Methods). For the computation of genetic correlations, we restricted the LD score regression intercept to zero as there is no sample overlap between the BSPC GWAS and the other GWASs.

other traits included in Table 1. Results for traits where the 95% confidence interval (CI) for the correlation excludes zero are in Table 2. For primary participation, the data errors that affect TNTC and WSPC are found to induce a bias in the $\chi^2$ statistics that is negatively correlated with the LD scores (Methods). While the described adjustments reduce the problem (Supplementary Note section 3; Methods), to avoid residual artefacts, the LDSC results here are based on BSPC only. For the other traits, GWAS results are based on the same unrelateds used for the results in Table 1 (Methods). While the primary participation genetic correlations with EA and the secondary participation traits are all significantly positive and substantial, they are all below 0.5. That could be partly because, while BSPC captures only direct genetic effects, the other GWASs were performed in the 'standard' manner and thus capture population effects that include both direct and nondirect genetic effects[9,10]. Genetic correlation estimates based on the adjusted WSPC results (Supplementary Table 2) are not very different. However, further research is needed to determine how best to use the WSPC and TNTC results for genetic correlation estimates.

Using the BSPC GWAS results, we obtained an estimate of 12.5% for the heritability of the liability score underlying primary participation (Methods). This required making several adjustments to the LDSC-estimated heritability and involved utilizing the 0.86 estimated value for the ratio $(F_{IBD2} - F_{IBD0})/(F_{samp} - f_{pop})$ (Supplementary Note section 7; Methods). For $\alpha = 0.055$, heritability of 12.5% would mean that the average value of the genetic component underlying the liability score is 0.72 s.d. higher for the participants than the population. Heritability estimates for the liability scores underlying the secondary participation events are distinctly smaller, ranging from 3.4% to 6.2% (Supplementary Table 3). This could be partly because these heritability estimates are computed from a biased sample. Notably, studying secondary participation is not the same as studying the bias underlying the primary sample, which affects everything that follows.

**Third principle of genetic induced ascertainment bias.** If genetics contribute to participation, there would be more close relative pairs among the participants than what is expected if participation is random. The third principle holds because if a participant has an above average genetic propensity to participate, so would its close relatives. Even though UKBB did not purposefully recruit families, the number of sib-pairs are twice as many as expected under random sampling[13]. It was speculated that mutual consultation and possibly shared environment contributed to correlated participation of close relatives[13].

Another likely contributor is shared DNA. With the liability-threshold model, $\alpha = 0.055$ and $\lambda_S = 2.0$ correspond to a 0.193 correlation between the liability scores of sib-pairs. Assuming the liability score heritability is 12.5%, the direct genetics effect can account for $(0.125/2)/0.193 = 32\%$ of the liability score correlation.

## Discussion

Participation bias, a concern for all sample surveys, is becoming increasingly relevant in the age of 'Big Data'[1,3]. In addition to obvious pitfalls, it could induce, for genetic studies, more subtle consequences that include collider bias[17] and the introduction of artificial epistatic effects (Supplementary Note section 8). Here, we show how ascertainment bias leaves footprints in the genetic data that can be exploited to study the bias itself. Our approach shares some principles with affected sib-pairs linkage analysis[22]. However, whereas the latter relies only on IBD, our method is an IBD-based association analysis.

Two of the three comparisons we proposed are sensitive to genotyping and phasing errors. This complicates analyses, but also creates a secondary usage of our method as a data-quality monitoring tool, like the Hardy–Weinberg equilibrium (HWE) test. For example, results that are significantly different between WSPC and BSPC at the MHC region would indicate data or data-processing problems. Furthermore, note that the genotyping-error induced bias also affects the transmission disequilibrium test—a test usually considered as fully robust.

There can be a common genetic component underlying many different participation events, for example, the pPGS constructed for primary participation associates with both the passive (being invited) and active (deciding to participate when invited) phases of the secondary participation events. Thus, the effect of a genetic component associated with participation could accumulate through many participation events of a person's lifespan, or it can be magnified through nested participation, for example, the participants in the dietary and physical activity studies have higher average genetic propensity to participate than the other UKBB participants, who have higher average propensities than the population. Instead of thinking of participation as a consequence of other characteristics and established traits, we propose that the propensity to participate in a wide range of events is a behavioral trait in its own right. While our method exploits information previously not utilized, even more could be achieved by combining it with other available information.

## Online content

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

# Methods

## Genetic data

The UKBB has a Research Tissue Bank approval (Research Ethics Committee reference 21/NW/0157) from the North West Multicenter Research Ethics Committee, and all participants gave informed consent. For our data application, we used the UKBB 500K data release previously described by Bycroft et al.[13]. We filtered out individuals that had withdrawn consent, were not in the kinship inference and phasing input, had a duplicate/twin in sample, with excess of third degree relatives, with missing rate above 2% as well as heterozygosity and missingness outliers and those who showed potential sex chromosome aneuploidy or potential sex mismatch.

The current analysis started with 658,565 biallelic sequence variants in the UKBB phased haplotype data[13]. We refer to them as SNPs even though a very small fraction are short indels. Compared with standard GWAS, some of our analyses are more sensitive to data artefacts. Thus, we applied a number of filters to the SNPs in addition to those applied by Bycroft et al.[13] (see Supplementary Note section 4 for details). Together with the trimming described below, association results were obtained for 500,632 high-quality SNPs.

## Identifying relatives and the group of unrelateds

Using kinship coefficients from the UKBB data release, we identified 16,668 White British sib-pairs (42.4% male, mean (year of birth (YOB)) = 1950.8), and 4,427 White British parent–offspring pairs (parents: 31.0% male, mean (YOB) = 1941.7; offsprings: 38.9% male, mean (YOB) = 1965.0, Supplementary Note section 1). White British descent was determined from self-identified ancestry and PC analysis[13]. For sibling-ships with more than two siblings, the first two participating siblings were chosen. Sib-pairs and parent–offspring pairs were chosen to have no overlap. Within the White British subset, 272,409 individuals have no relatives within the UKBB of third degree or closer (46.7% male, mean (YOB) = 1951.3). Those are referred to as the 'unrelateds' here.

## Inferring IBD segments and trimming

We inferred IBD segments for sibling pairs using snipar[9], a program designed for sib-pairs. Compared with KING[18] (v.2.2.4) a program not tailored for sib-pairs, snipar entails a substantial improvement in IBD estimation (Extended Data Fig. 2). Neither program uses phasing information as input. Considering the uncertainty around recombination events, we trimmed away 250 SNPs from the beginnings and ends of the inferred IBD segments before tabulating allele frequencies (Extended Data Fig. 3). This removed 11,000 SNPs all together ($2 \times 250 \times 22$) and reduced the sample sizes for the remaining SNPs (median reduction of 1,565 sibling pairs).

## Inferring shared allele

For a biallelic SNP and a relative pair, there are five possible unordered combinations of genotypes for IBD1. The IBD shared allele is clear except when both individuals are heterozygous. For that, the phasing information provided[13] was utilized. The closest neighboring SNP, for which one individual was heterozygous and the other homozygous, was identified and used to determine the shared haplotype and, through that, the shared allele of the target SNP (Extended Data Fig. 4).

## Test statistics

For each SNP, we computed separate $t$-statistics for TNTC, WSPC and BSPC, by dividing each of the frequency differences, equations (1) to (3), by its s.e. For TNTC and WSPC, the s.e. values were computed assuming the shared versus not-shared frequency differences of each pair were independent of each other. For BSPC, for every SNP, the allele frequencies computed for individual sibling pairs, whether IBD0 or IBD2 pairs, were assumed to be independent of each other. No further

assumptions, for example, HWE or independence of genotypes of two siblings in the IBD0 case, were made. Essentially, TNTC and WSPC were treated as one-sample $t$-tests (of differences), and BSPC were treated as a two-sample $t$-test. Specific equations are given in Supplementary Note section 2, which also contains information on sample sizes.

## Sources of errors inducing the major-allele bias in the test statistics

When the initial results for all the SNPs were examined together, the $t$-statistics showed a tendency towards favouring the major allele as having higher frequency in the shared alleles. We identified three sources to this bias:

**IBD estimation errors.** For every sib-pair, the analysis starts with determining the IBD status for each SNP. Error here could lead to biases. This problem was addressed by replacing the KING[18] program by snipar[9], as noted above. Together with the trimming noted above, the impact of IBD estimation errors seems to be mostly eliminated (see Supplementary Note section 3 for further discussion).

**Phasing errors.** Phasing errors can induce a systematic major-allele bias in the WSPC and TNTC $t$-statistics. We focus more on WSPC below as it captures essential the same real effects as BSPC. The induced bias on TNTC is similar.

For TNTC and WSPC, when the genotypes of both relatives are heterozygous, to determine the shared allele, we use phased haplotypes that include neighboring SNPs. Phasing errors can thus induce a bias. Let the frequency of allele 1 be $f_{pop} = f$. For simplicity, assume the SNP is in HWE in the population and not associated with participation. When two siblings share one allele IBD and both are heterozygous, the chance that the shared allele is 1 is $(1 - f)$. Let $\varepsilon_f$, a function of $f$, be the error rate of calling the shared allele 0 when it is actually 1. By symmetry, the error rate of calling the shared allele 1 when it is actually 0 is $\varepsilon_{1-f}$. Because the probability of the shared allele being actually 1 is $(1 - f)$, the induced bias of the two type of errors combined is $f\varepsilon_{1-f} - (1 - f)\varepsilon_f$. For $f > 0.5$, the bias is positive if

$$\frac{\varepsilon_f}{\varepsilon_{1-f}} < \frac{f}{1-f} \qquad (8)$$

Using data from 739 UKBB trios, where the shared allele between a parent–offspring pair in the double-heterozygotes case could be resolved by the genotype of the other parent, we estimated $\varepsilon_f$ (Extended Data Fig. 5), which satisfies equation (8). More details about the quantitative results and the induced bias on the $t$-statistics are given in Supplementary Note section 3. One observation, highlighted by Extended Data Fig. 6, is that, for SNPs with MAF > 0.10, around 50% of the observed bias of the WSPC $t$-statistics can be explained by the double-heterozygote errors. However, as MAF becomes small, the fraction of the empirical bias explainable by double-heterozygote errors decreases, for example, to 21% when MAF = 0.01. We believe genotype-calling errors are responsible for the additional bias.

**Genotyping errors.** Genotyping errors can induce a systemic bias to the WSPC and TNTC statistics. Simulations show that random errors where each genotype has a small probability to be replaced by a random genotype drawn from the population would induce a bias on $F_{IBD1S} - F_{IBD1NS}$ which is positive if allele 1 has frequency >0.5. This happens even though such error mechanism would not even change the sampling distribution of the called genotypes. Furthermore, genotype error rate is known to be higher for SNPs with lower MAFs. We believe the main driving force of the major-allele bias is the minor allele being overcalled in the not-shared alleles (explanation in Supplementary Note section 3).

## Two-step adjustment: adjustments of $t$-statistics and $\chi^2$ statistics

For WSPC and TNTC, separately, we made a simple adjustment by regressing the $t$-statistics on centred allele frequency ($cf$), $cf = (f - 0.5)$, through the origin and took the residuals as the adjusted values. This corresponds to deducting $(0.6464 \times cf)$ and $(0.3781 \times cf)$ from the unadjusted $t$-statistics of WSPC and TNTC, respectively. This adjustment avoids artificial association with other GWASs that are also subject to a major-allele bias, but resulting from a different mechanism. If the other GWASs exhibit major-allele biases for the same reasons, this adjustment would reduce, but not eliminate, an artificial association.

**Effect of the major-allele bias on the $\chi^2$ statistics.** The errors underlying the major-allele bias of the $t$-statistics would also inflate the average values of the $\chi^2$ statistics. The $t$-statistic adjustment above will reduce, but not eliminate, this inflation. This is because the adjustment is based on allele frequency, that is, it removes the average bias of alleles with similar frequency. As SNPs with similar allele frequency will have biases that vary around the average, the variation of the adjusted $t$-statistics would still be inflated, and through that inflate the average value of the $\chi^2$ statistics. Notably, the average $\chi^2$ value for BSPC is 1.011 for the 500,632 SNPs, 1.025 for the 110,533 SNPs with MAF > 0.25, and 1.007 for the 390,099 SNPs with MAF < 0.25. By contrast, for WSPC, the corresponding average $\chi^2$ values are 1.136, 1.039 and 1.163, respectively, without $t$-statistic adjustment, and 1.074, 1.029 and 1.086, respectively, after adjustment. With $t$-statistic adjustment, the average $\chi^2$ value for WSPC is only modestly inflated relative to BSPC for SNPs with MAF > 0.25, but remain substantially inflated for SNPs with MAF < 0.25. Moreover, while the $\chi^2$ statistics for BSCP are slightly positively correlated with MAF ($r = 0.006$), the $\chi^2$ statistics for WSCP, even with $t$-statistic adjustment, have a larger but negative correlation with MAF ($r = -0.021$). The latter is because the major-allele bias of an SNP increases as MAF decreases. The negative correlation between the WSPC $\chi^2$ values and MAF is problematic for the application of LD score regression as the LD scores have a substantial positive correlation ($r = 0.35$) with MAF. Without further adjustments, applying LD score regression to the WSPC $\chi^2$ values gives a negative fitted slope, or a negative estimated heritability. To address this, we performed MAF-specific genomic control. Specifically, we started by regressing the $\chi^2$ values computed from the adjusted $t$-statistics on polynomial of MAF up to the third power. The fitted values of the $\chi^2$ values as a function of MAF are displayed in Extended Data Fig. 7. The $\chi^2$ values were then adjusted by dividing the original values by the fitted values. By construction, these $\chi^2$ values have average equal to 1. Taking the square-root and multiplying by the sign of the adjusted $t$-statistics gave the 'final' $z$-(or $t$-) scores of the WSPC GWAS. The same method was used for the TNTC GWAS. These $z$-scores were used to evaluate the significance of individual SNPs and to compute the polygenic scores used for Table 1. These adjustments reduce the impact of the artefacts, but the results are still imperfect (see Supplementary Note section 3 for further discussion). Thus, the LD score regression estimates of genetic correlations in Table 2 of the main text are based on the BSPC results only, and so is the heritability estimate given. However, the estimates in Supplementary Table 2, based on the adjusted WSPC results, are broadly consistent with those in Table 2, supporting that the adjustments have reduced the problems that arise in the application of LD score regression.

When we use the GWAS results to construct polygenic scores, we find the adjustments described above to have a very small effect on the predictive power of the polygenic scores. In particular, the polygenic score constructed from the WSPC $t$-statistics, with or without adjustments, has very similar predictive power as the polygenic score constructed from the BSPC $t$-statistics (Table 1 and Supplementary Table 1). This is because the bias is quite small per SNP for this filtered set and so would only be adding a little noise to the polygenic score prediction.

## Polygenic score analysis

The pPGS was computed with PLINK 1.90 (ref. 24) by summing over the weighted genotypes of the 500,632 SNPs fulfilling quality control, using the $z$- (or $t$-) statistics from the primary participation GWASs as weights. The relationship between the pPGS, standardized to have a variance of 1, and the phenotypes in Table 1 was estimated with a linear regression and logistic regression in R (v.3.4.3) in the group of White British unrelateds, taking sex, YOB, age at recruitment, genotyping array and 40 PCs into account. The quantitative phenotypes were transformed to have a variance of 1 for men and women separately (for further information see Supplementary Note section 5). To account for population stratification, the $P$ value for each pPGS-phenotype association was adjusted through dividing the squared test statistic ($t$-test for quantitative phenotypes and $z$-test for binary phenotypes) by the LD score regression intercept estimated from GWAS summary statistics for the corresponding phenotype (see below).

## LD score regression

We performed LD score regression with the program LDSC (v.1.0.1) (ref. 19) using the European Ancestry LD scores computed by the Pan-UKB team[23] (downloaded on 7 April 2021). Analyses were based on the 500,632 SNPs used to compute the pPGS.

LD score regression intercepts and genetic correlations were estimated for the primary participation test statistics described above and the phenotypes shown in Table 1. We obtained summary statistics for the phenotypes shown in Table 1 by running GWASs in the group of White British unrelated individuals in the UKBB using BOLT-LMM (v.2.3) (ref. 25). For quantitative phenotypes, which had been adjusted for sex, age, YOB and 40 PCs and transformed to have variance of 1 as described in Supplementary Note section 5, genotyping array was added as an additional covariate in the GWASs. For the binary phenotypes, YOB, age at recruitment up to the order of three, 40 PCs, sex and genotyping array were added as covariates. For LD score regression, we used the standard linear regression $P$ values, P_LINREG, from the BOLT output. Heritability of participation traits was estimated for a liability-threshold model (see below).

## Liability-threshold model for participation of individuals and sib-pairs

The liability-threshold model assumes that a liability score, denoted here by $X$, underlies a 0/1 trait or response. $X$ is assumed to have a standard normal distribution (or roughly so). Let $I$ be the 0/1 participation variable, and participation rate is $P(I = 1) = \alpha$. It is assumed that $I = 1$ if $X > \tau$, where $\tau = \Phi^{-1}(1 - \alpha)$ and $\Phi$ is the cumulative distribution of the standard normal. Correlation of participation between individuals is modeled through the correlation of their liability scores.

For $n$ sib-pairs, let $i = 1, \ldots, n$ index the pairs and $j = 1, 2$ index the two siblings in a pair. Focusing on one SNP with its standardized genotype denoted by $g$, the liability of sibling $ij$ is modeled as

$$X_{ij} = w_1 g_{ij} + w_A A_i + w_B B_{ij} \tag{9}$$

where $A_i$ and $B_{ij}$ are standard normal variables. The variables $A_i$, $B_{i1}$ and $B_{i2}$ are assumed to be independent of $g_{i1}$ and $g_{i2}$, and each other. $A_i$ captures effects from shared environment as well as shared genetic factors other than $g$. We assume $w_1^2 + w_A^2 + w_B^2 = 1$ so that $\mathrm{var}(X_{ij}) = 1$. Because $\mathrm{cor}(g_{i1}, g_{i2}) = 1/2$, $\mathrm{cor}(X_{i1}, X_{i2}) = w_A^2 + w_1^2/2$. Here $X_{ij}$ is not exactly normally distributed because of $g_{ij}$, but it is close if $w_1$ is small.

## Simulations to study relationships between various frequency differences of an SNP.

Results in Fig. 4 were simulated with $\alpha = 0.055$, the participation rate of UKBB. Allele 1 of the SNP is assumed to have a

population frequency of 0.5 and a positive participation effect with $w_1^2 = 0.001$. Sixteen different values of $w_A$ are chosen so that $\mathrm{cor}(X_{i1}, X_{i2}) = w_A^2 + w_1^2/2$ takes on values of 0.0005, 0.025, 0.050, 0.075, 0.100, 0.125, 0.150, 0.175, 0.193, 0.225, 0.250, 0.275, 0.300, 0.325, 0.350 and 0.375. Notably, $\mathrm{cor}(X_{i1}, X_{i2}) = 0.193$ leads to $\lambda_S = 2.0$, the reported sibling enrichment in UKBB[13].

Simulations for dominant and recessive models were performed similarly. For the dominant model, the $g_{ij}$ in the liability score definition is taken as the standardized version of a 0/1 variable, which is 1 if the actual genotype is 1 or 2, and 0 otherwise. For the recessive model, $g_{ij}$ is the standardized version of a 0/1 variable, which is 1 if the actual genotype is 2.

### Estimating heritability for liability scores

Here, we show how the heritability of the participation traits were estimated, starting with secondary participation. When the LDSC program is given the $\chi^2$ statistics and sample sizes for a set of genome-wide SNPs, the heritability estimate produced is an estimate of the fraction of variance of the trait accountable by the genetic component, or $r^2$ between trait and the genetic component. If genetic component $G$ influences the participation variable $I$ through liability score $X$, then

$$\mathrm{cor}(G, I) = \mathrm{cor}(G, X) \times \mathrm{cor}(X, I) \tag{10}$$

Heritability of $X$ and $I$ thus satisfy

$$h^2(X) = \frac{h^2(I)}{\mathrm{cor}^2(X, I)} \tag{11}$$

with

$$\mathrm{cor}(X, I) = \frac{E(XI) - E(X)E(I)}{\sqrt{(\mathrm{var}(X)\mathrm{var}(I))}} = \frac{\alpha E(X|I=1)}{\sqrt{\alpha(1-\alpha)}} = \frac{\alpha \frac{\phi(\tau)}{1-\Phi(\tau)}}{\sqrt{\alpha(1-\alpha)}} = \frac{\phi(\tau)}{\sqrt{\alpha(1-\alpha)}} \tag{12}$$

where $\phi$ is the density function of the standard normal and $E$ stands for expectation. Consider the Dietary Study invitation event. Feeding the LDSC program the $\chi^2$ statistics from the GWAS analyses and a sample size of 272,409 (166,993 invited; 105,416 not-invited), the heritability estimate is 0.0372. Here $\alpha = 166993/272409 = 0.613$, $\tau = \Phi^{-1}(1 - 0.613) = -0.287$, and $\mathrm{cor}(X, I)$ is 0.786. The heritability of $X$ is estimated as the estimated heritability of $I$ multiplied by the adjustment factor $[1/\mathrm{cor}^2(X, I)]$, or

$$0.0372 \times \frac{1}{0.786^2} = 0.060 \tag{13}$$

The heritability of the liability scores of the other secondary participation events are similarly estimated. The $r^2$ between a genetic component, or the genotype of an individual SNP, is weaker, or statistically less efficient, with $I$ than with $X$. In this sense, the adjustment factor is inversely proportional to the statistical efficiencies of the test statistics used, a principle that also applies to the two other adjustments described below. Moreover, Supplementary Note section 7 describes how the adjustment could be alternatively applied through providing the LDSC program with modified sample sizes.

With primary participation, heritability estimation requires the adjustment step above plus two others because the BSPC results are not direct comparisons of genotypes of participants and nonparticipants. Let $n_{\mathrm{IBD2}}$ and $n_{\mathrm{IBD0}}$ be the respective number of IBD2 and IBD0 pairs at the SNP location. Feeding the LDSC program the $\chi^2$ statistics from the BSPC GWAS with number of 'cases' equals $n_{\mathrm{IBD2}}$ and number of controls equals $2 \times n_{\mathrm{IBD0}}$, the estimated heritability given is 0.0838.

Here $\alpha = 0.055$, $\tau = 1.598$ and $\mathrm{cor}(X, I) = 0.488$. Hence, the first adjustment factor is $(1/0.488)^2 = 4.2$. However, the numbers of participants and nonparticipants have a 0.055 to 0.945 ratio. By contrast, the cases to controls ratio for BSPC is around 1:2, much more efficient as the variance of the comparison is roughly proportional to

$$\frac{1}{n_{\mathrm{cases}}} + \frac{1}{n_{\mathrm{controls}}} \tag{14}$$

As variance here is inversely proportional to efficiency, this leads to the adjustment factor, for arbitrary $n$,

$$\frac{\frac{1}{n \times (1/3)} + \frac{1}{n \times (2/3)}}{\frac{1}{n \times (0.055)} + \frac{1}{n \times (0.945)}} = \frac{3 + \left(\frac{3}{2}\right)}{18.18 + 1.06} = 0.2338 \tag{15}$$

There is a third adjustment because the allele frequency difference between the IBD2 sibs and the IBD0 sibs is smaller than the frequency difference between the participants and nonparticipants. Specifically, for $\alpha = 0.055$ and $\lambda_S = 2.0$, our simulations gave

$$E\left[\frac{F_{\mathrm{IBD2}} - F_{\mathrm{IBD0}}}{F_{\mathrm{samp}} - f_{\mathrm{pop}}}\right] \approx 0.86 \tag{16}$$

Since $(F_{\mathrm{samp}} - f_{\mathrm{pop}}) = (1 - \alpha)(F_{\mathrm{samp}} - F_{\mathrm{nonparticipants}})$

$$E\left[\frac{F_{\mathrm{IBD2}} - F_{\mathrm{IBD0}}}{F_{\mathrm{samp}} - F_{\mathrm{nonparticipants}}}\right] \approx 0.86 \times (1 - \alpha) = 0.86 \times 0.945 = 0.8127 \tag{17}$$

Because efficiency is proportional to effect$^2$, the adjustment factor is

$$\left(\frac{1}{0.8127}\right)^2 = 1.514 \tag{18}$$

With the three adjustments, the estimated heritability of primary participation is

$$0.0838 \times 4.2 \times 0.2338 \times 1.514 = 0.125 \tag{19}$$

or 12.5%.

### Reporting summary

Further information on research design is available in the Nature Portfolio Reporting Summary linked to this article.

## Data availability

The primary participation GWAS summary statistics generated in this study have been deposited to the GWAS catalog under the accession codes GCST90267220, GCST90267221, GCST90267222 and GCST90267223. Researchers can apply for access to individual-level UKBB data on their website (http://www.ukbiobank.ac.uk/register-apply/).

## Code availability

The genotype data was handled with QCTOOL v.2.0.1 (https://www.well.ox.ac.uk/~gav/qctool_v2/) and PLINK[24] v.1.90 (https://www.cog-genomics.org/plink/1.9/) and v.2.00 (https://www.cog-genomics.org/plink/2.0/). IBD segments of siblings were inferred with snipar[9] (https://github.com/AlexTISYoung/snipar/blob/ff48c642da1e45067a-fae1e21f5e5e450d4d4ef9/) and also with KING[18] v.2.2.4 (https://www.kingrelatedness.com) for comparison. Statistical analysis were performed in Python v.2.7.11 (https://www.python.org) and R v.3.4.3 (https://www.R-project.org/). LD score regression intercepts and estimates of heritability and genetic correlations were attained with LDSC[19,21] v.1.0.1 (https://github.com/bulik/ldsc). GWAS summary statistics for the phenotypes shown in Table 1 were attained with BOLT-LMM[25]

v.2.3 (https://alkesgroup.broadinstitute.org/BOLT-LMM/). Scripts for reproducing the analysis in the current study are available at https://github.com/stefaniabe/PrimaryParticipationGWAS (ref. 26).

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

## Acknowledgements

We thank S. Song, A. Young, C. Lindgren, M. Przeworski, D. Palmer, M. Mills and T. Ferreira for valuable discussions about this study. A.K. is supported by the Li Ka Shing Foundation and the Leverhulme Trust (Grant RC-2018-003). S.B. is supported by the Li Ka Shing Foundation and the Goodger and Schorstein scholarship. This research has been conducted using the UKBB Resource under application number 11867. The computational aspects of this research were supported by the Wellcome Trust Core Award Grant Number 203141/Z/16/Z and the NIHR Oxford BRC. The views expressed are those of the authors and not necessarily those of the NHS, the NIHR or the Department of Health. The funders had no role in study design, data collection and analysis, decision to publish or preparation of the manuscript.

## Author contributions

A.K. developed the study concept. A.K. and S.B. designed the study and the methodology, performed the investigation and data application, and wrote, reviewed and edited the manuscript.

## Competing interests

The authors declare no competing interests.

## Additional information

**Extended data** is available for this paper at https://doi.org/10.1038/s41588-023-01439-2.

**Correspondence and requests for materials** should be addressed to Stefania Benonisdottir or Augustine Kong.

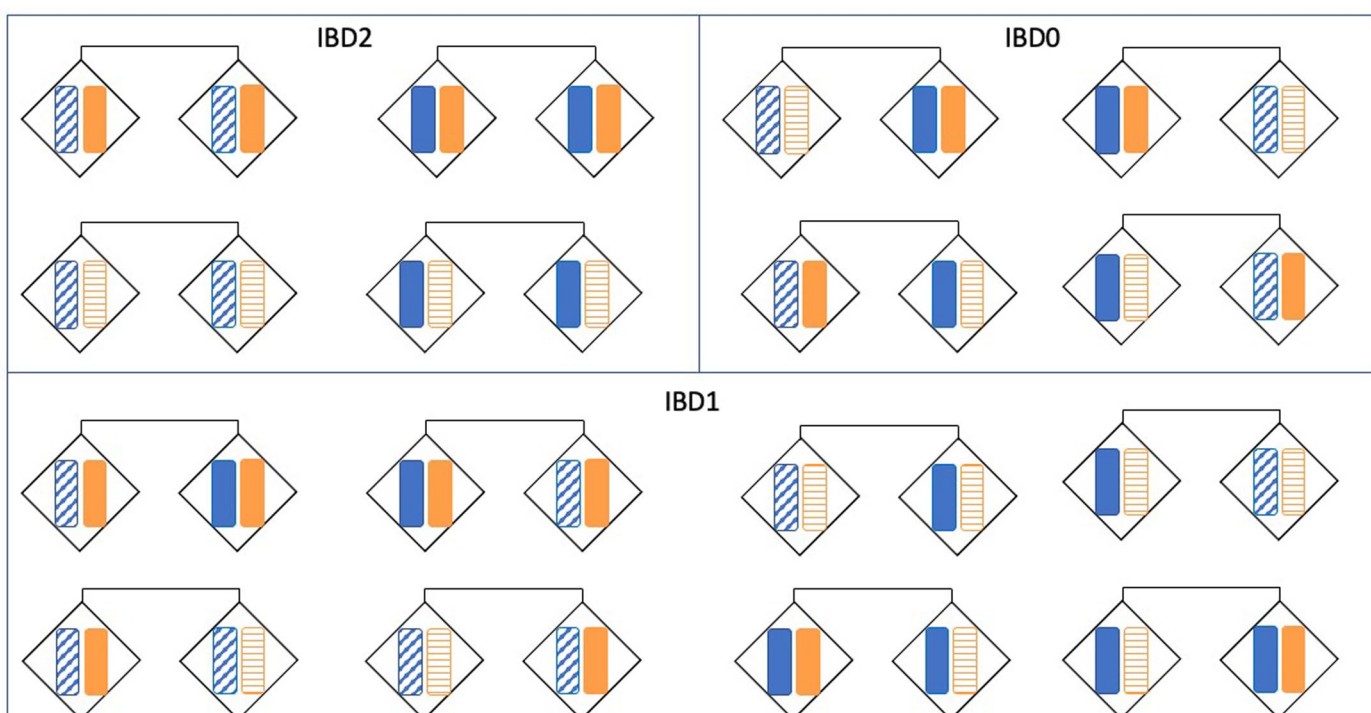

**Extended Data Fig. 1 | Parental transmissions to sibling pairs.** Displayed are the 16 possible equally likely combinations of transmissions of parental genetic segments to a sibling pair at a given locus. The father has two blue genetic segments, one solid and one striped, and the mother has two orange genetic segments, solid and striped. The different colors and fills indicate distinct origins of inheritance. That is, the four parental genetic segments could be identical by state but they are all distinct with regard to grandparental origin. The sibling pairs are shown as diamond shapes, each carrying one blue genetic segment (solid or striped) inherited from the father and one orange genetic segment (solid or striped) inherited from the mother. In the case the siblings share one segment IBD (IBD1), 8 out of the 16 combinations, the shared segment is paternal for 4 combinations and maternal for the other 4 combinations.

a) *KING*

b) *snipar*

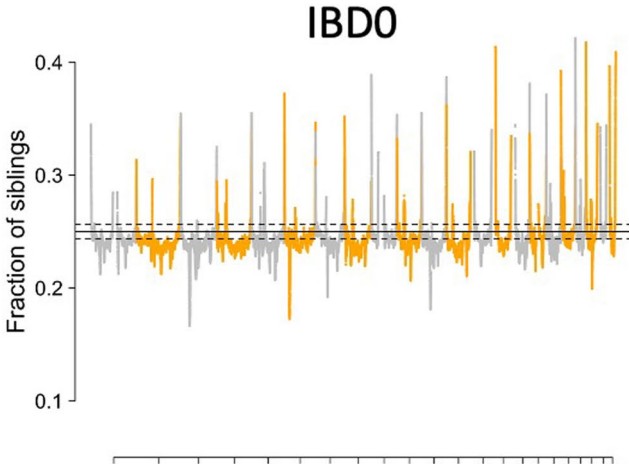

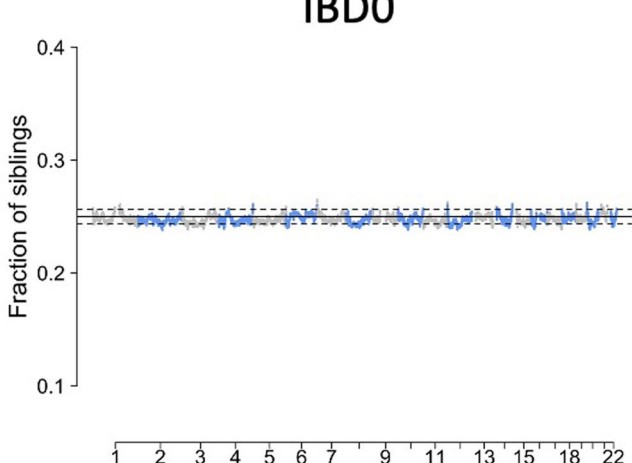

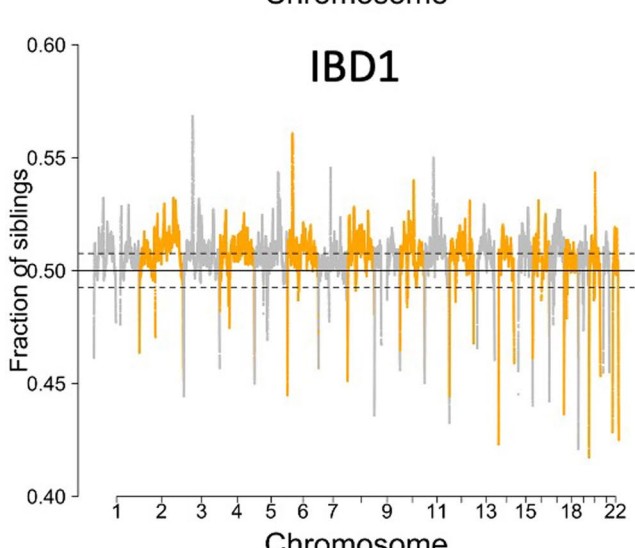

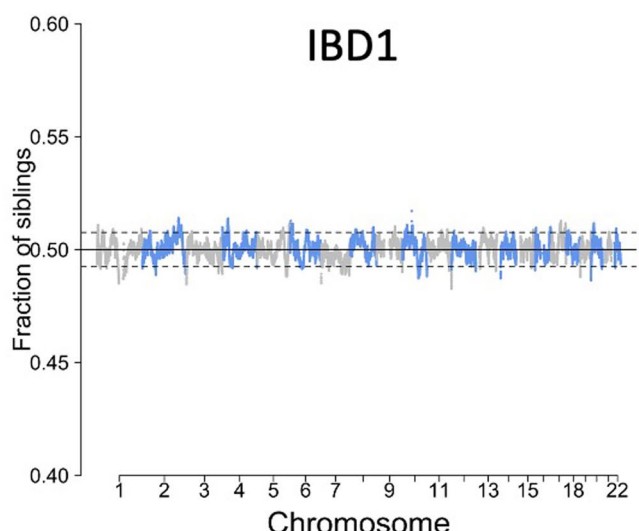

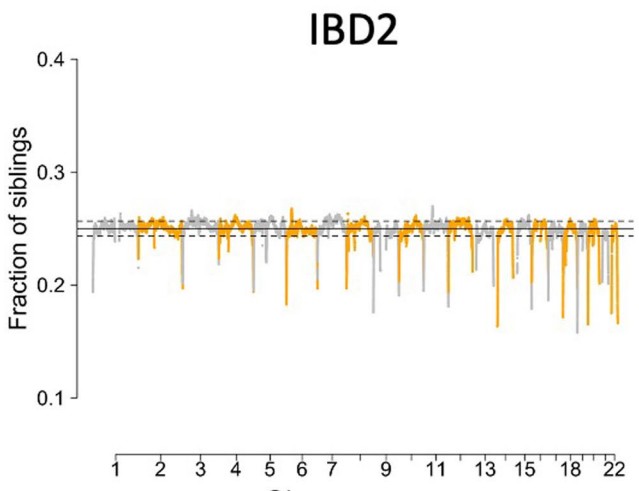

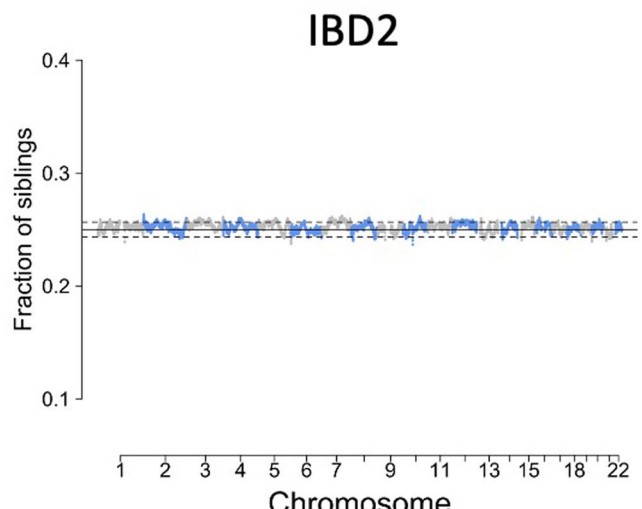

**Extended Data Fig. 2 | See next page for caption.**

**Extended Data Fig. 2 | Expected and called fractions of sibling pairs sharing 0, 1, and 2 alleles IBD.** The figure shows, for each SNP, chromosomal position (x-axis, build 37) and the estimated IBD fractions among the 16,668 white British sibling pairs in UKBB (y-axis). The black solid lines denote the theoretical expected fraction for each IBD state, equals 0.25, 0.5, and 0.25 for IBD0, IBD1, and IBD2 respectively. The two black dashed lines indicate the theoretical 95% probability interval, *that is* expectation ± 1.96 SD. Figure a) shows the empirical sibling fractions for each of the three IBD states computed based on results from the program *KING*[18], and figure b) shows the sibling fractions computed with the results from the program *snipar*[9].

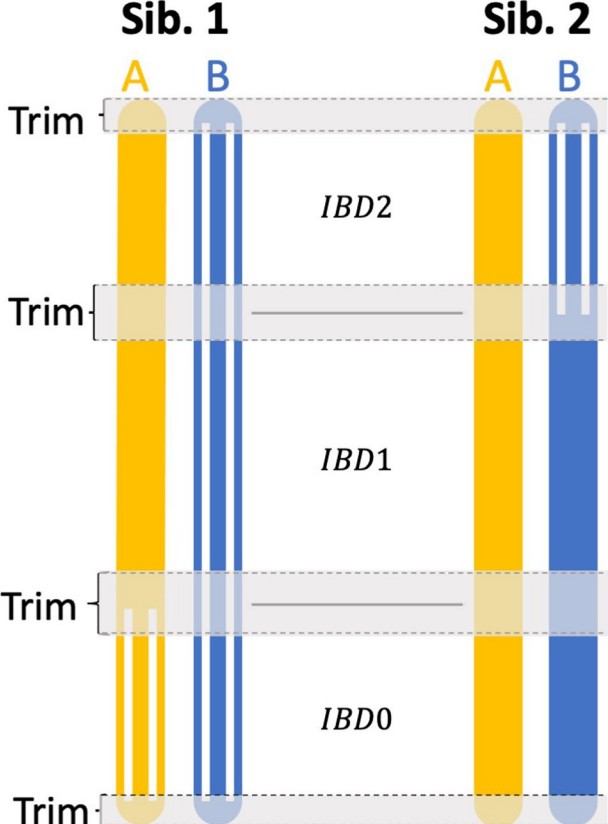

**Extended Data Fig. 3 | Trimming SNPs at the beginning and end of IBD regions.** Noting that the error-rate of inferring IBD state is higher in the beginning and end of IBD regions, we trimmed away 250 SNPs from the beginning and end of each called IBD segment.

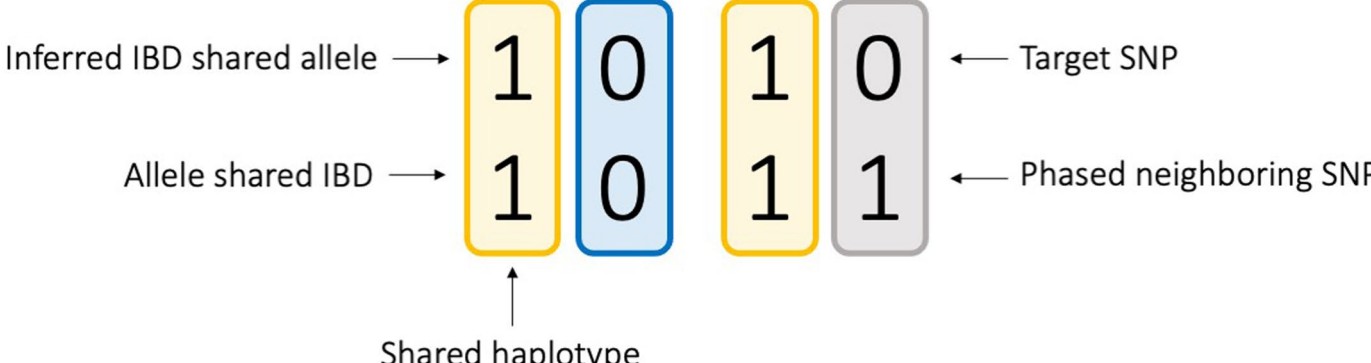

**Extended Data Fig. 4 | Inferring shared allele for IBD1 when both individuals are heterozygous for target SNP.** Within the IBD1 region, we search for a neighboring SNP for which one individual is heterozygous while the other is homozygous. If such a neighboring SNP exists, and is phased with the target SNP, the shared allele of the target SNP can be inferred through the shared haplotype. This method was also used by Young et al.[9] to infer the IBD1 shared allele.

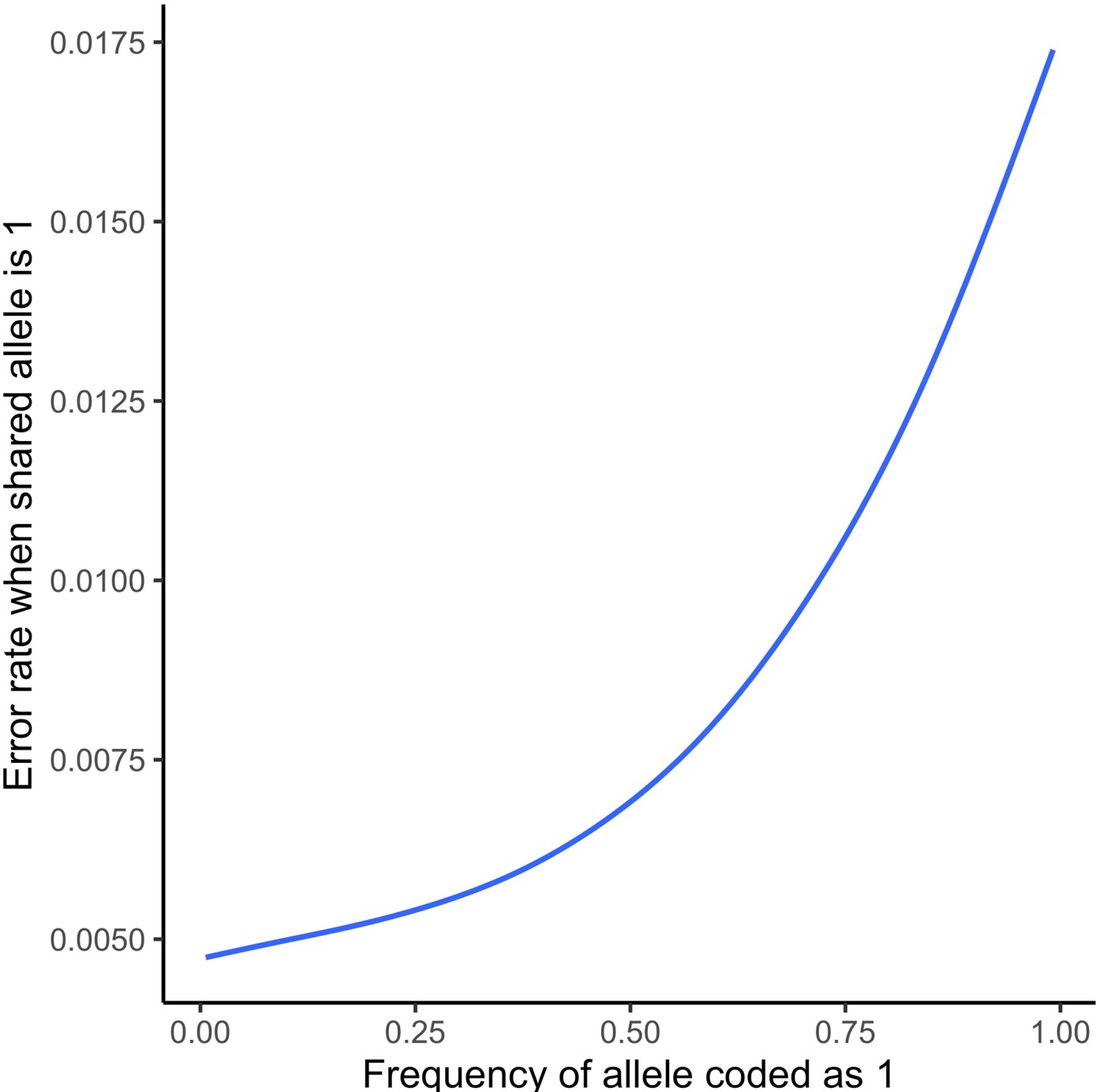

**Extended Data Fig. 5 | Phasing error rate as a function of allele frequency.** For each biallelic sequence variant, we estimate the phasing error rate (y-axis) from trios where the offspring and one parent are heterozygous while the other parent is homozygous. The latter allowed us to determine the shared allele without using phasing and is taken as the truth. Error is when the shared allele deduced through phasing for the double-heterozygotes parent–offspring pair differs from the 'truth' supported by the genotype of the homozygous parent. The error rate here is, for instances where the true shared allele is 1, the fraction of times that allele 0 is deduced as the shared allele through phasing. The solid line shows the fit from regressing the estimated error rate on allele frequency up to the third power in the set of 500,632 SNPs.

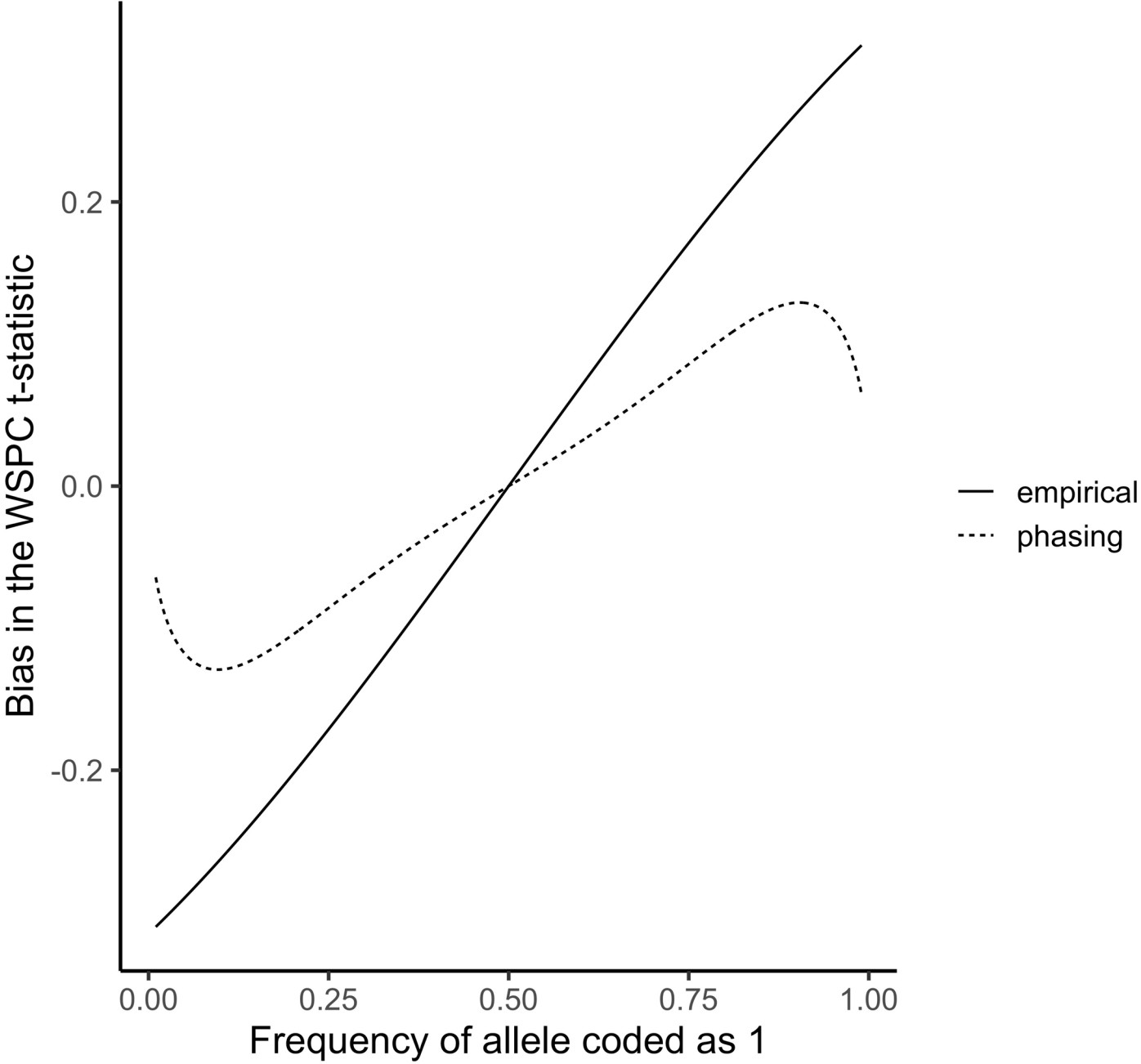

**Extended Data Fig. 6 | Estimated bias of the WSPC t-statistics as a function of allele frequency.** The solid line shows the fit from regressing the unadjusted WSPC t-statistics through the origin on centered allele frequency ($cf = f - 0.5$) and $cf^3$ in the set of 500,632 SNPs. The dashed line shows the fit for the estimated bias induced by miscalling the shared allele for the double-heterozygotes as a function of allele frequency. As described in Supplementary Note section 3, the estimated phasing induced bias was computed as $2f(1-f) [f\epsilon_{1-f} - (1-f)\epsilon_f]/SE_f$ with $f$ being the frequency for the allele coded as 1, $\epsilon_{1-f}$ and $\epsilon_f$ being the estimated phasing error rate for a given $f$ (see Extended Data Fig. 5) and $SE_f$ being the standard error of the shared-not-shared allele frequency difference for a given $f$. We note that, mainly due to the variation in sample sizes, $SE_f$ has modest variation among SNPs with very similar $f$. For the figure here, a fitted value of $SE_f$ is used.

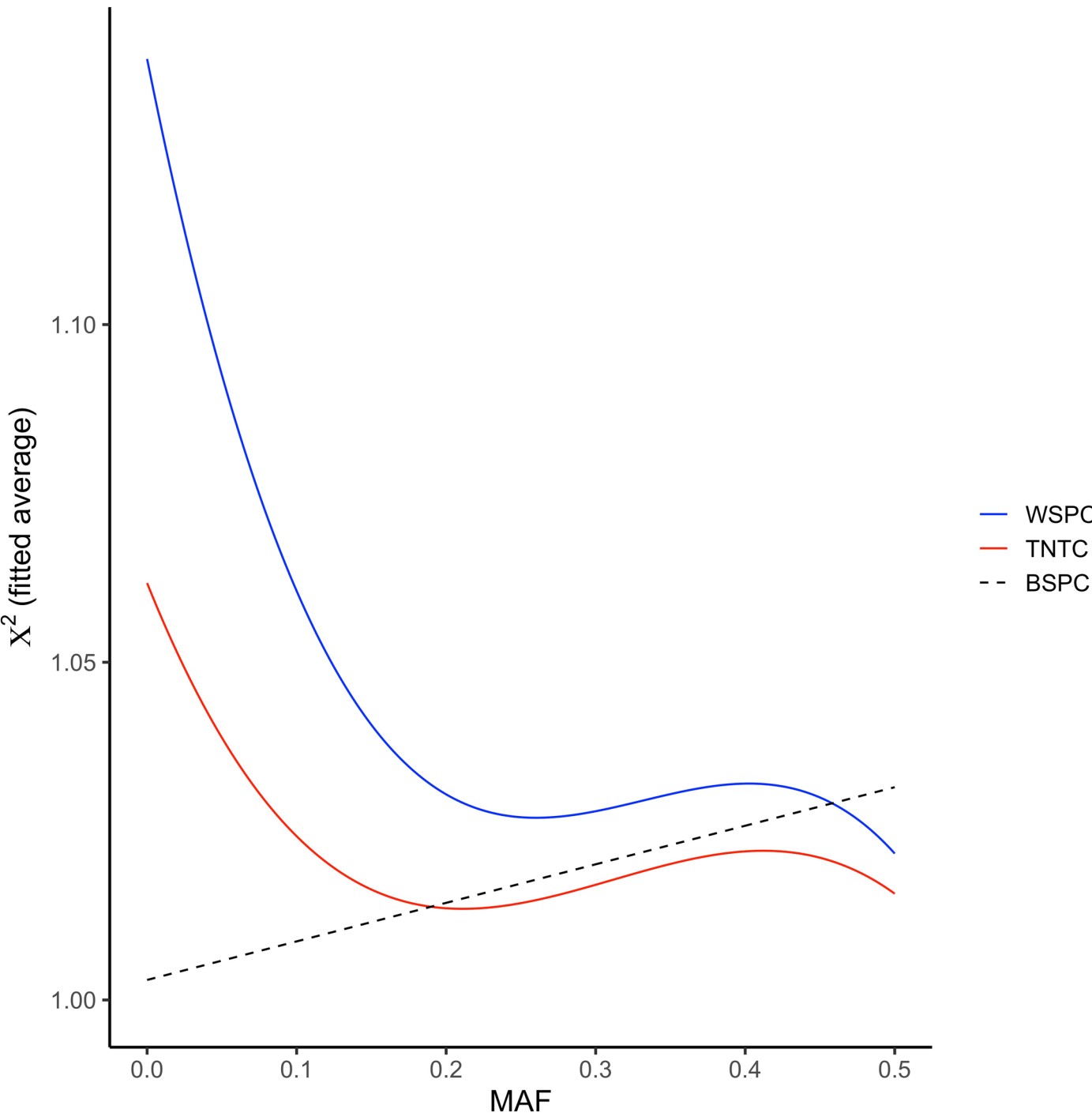

**Extended Data Fig. 7 | $\chi^2$ values as a function of minor allele frequency (*MAF*).** The two solid lines show the fit from regressing the $\chi^2$ statistics, computed from the allele-frequency adjusted TNTC and WSPC t-statistics, on MAF up to the third power. The t-statistics are for the 500,632 SNPs. The fitted value for a particular MAF can be interpreted as the average $\chi^2$ values for SNPs with MAFs close to that. The broken line is the corresponding fit for the BSPC $\chi^2$ statistics. Given that BSPC and WSPC capture similar true effects with comparable power, the difference between the MAF-specific fitted/average $\chi^2$ values is a measure of the average inflation of the WSPC $\chi^2$ values. Notably, for WSPC, the fitted $\chi^2$ value is much higher for SNPs with low MAFs. By contrast, the fitted $\chi^2$ value for BSPC has an increasing trend as MAF gets bigger. When MAF is low, the WSPC fitted $\chi^2$ value is substantially higher than that of BSPC, indicating that data errors are inducing a higher inflation there. As MAF increases, the difference between the WSPC and BSPC fitted $\chi^2$ values decreases. The fitted $\chi^2$ value of BSPC actually becomes slightly bigger than that of WSPC for MAF > 0.46, although that difference is not statistically significant. This is consistent with the WSPC results being close to unbiased when MAF is close to 0.5, which makes sense as the difference between major and minor alleles is small, and so is the major allele effect, when MAF is close to 0.5. The TNTC fitted $\chi^2$ value is in general smaller than that of WSPC. That is mainly due to the smaller effective sample size of TNTC, which affects the contributions of both the true effect and the bias to the $\chi^2$ statistics.

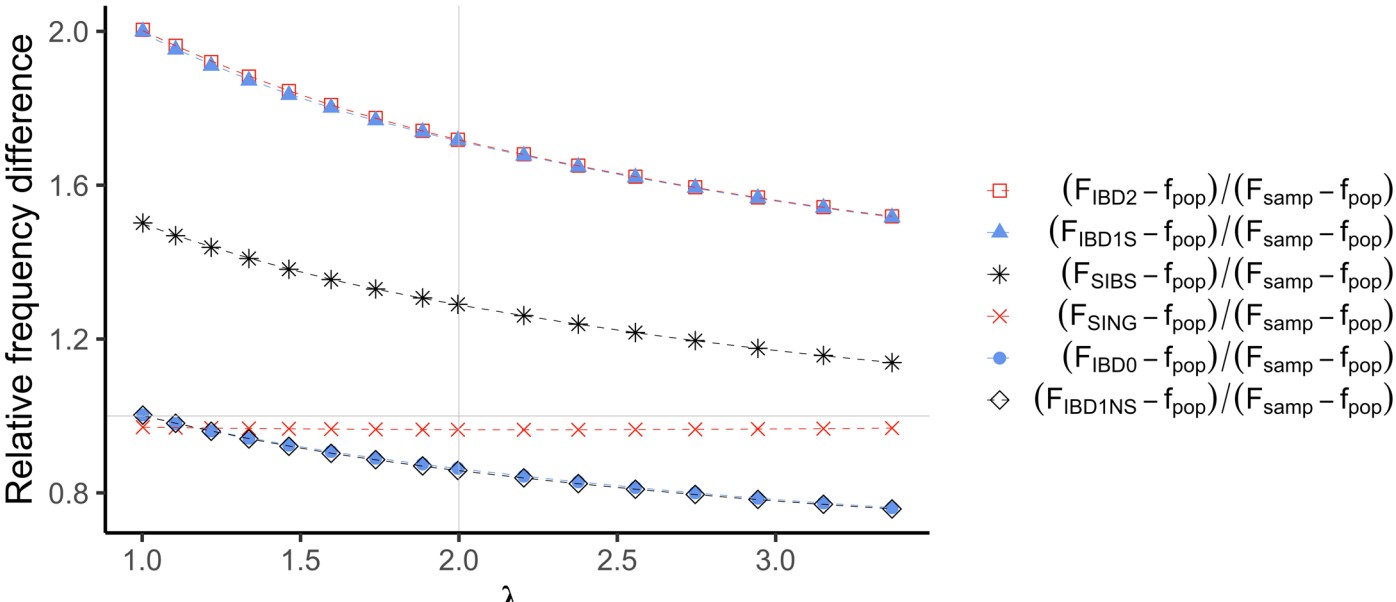

**Extended Data Fig. 8 | Relative frequency differences as a function of enrichment of sibling pairs in sample.** Displayed are relative allele frequency differences for different groups and segments as functions of the sibling recurrence participation ratio, $\lambda_S$. These differences are estimated from the same simulations underlying Fig. 4 and are described in the main text and Methods. $F_{IBD2}$ and $F_{IBD0}$ denote the allele frequency among sibling pairs sharing the SNP IBD2 and IBD0 respectively, while $F_{IBD1S}$ and $F_{IBD1NS}$ denote the allele frequency among the shared and not-shared alleles among sibling pairs sharing the SNP IBD1. $F_{SIBS}$ is the allele frequency in the participating sibling pairs, $F_{SING}$ is the allele frequency in the participating individuals whose sibling does not participate and $f_{pop}$ is the population allele frequency.

Augustine Kong

# Reporting Summary

## Statistics

For all statistical analyses, confirm that the following items are present in the figure legend, table legend, main text, or Methods section.

| n/a | Confirmed | |
|---|---|---|
| ☐ | ☒ | The exact sample size (*n*) for each experimental group/condition, given as a discrete number and unit of measurement |
| ☒ | ☐ | A statement on whether measurements were taken from distinct samples or whether the same sample was measured repeatedly |
| ☐ | ☒ | The statistical test(s) used AND whether they are one- or two-sided<br>*Only common tests should be described solely by name; describe more complex techniques in the Methods section.* |
| ☐ | ☒ | A description of all covariates tested |
| ☐ | ☒ | A description of any assumptions or corrections, such as tests of normality and adjustment for multiple comparisons |
| ☐ | ☒ | A full description of the statistical parameters including central tendency (e.g. means) or other basic estimates (e.g. regression coefficient) AND variation (e.g. standard deviation) or associated estimates of uncertainty (e.g. confidence intervals) |
| ☐ | ☒ | For null hypothesis testing, the test statistic (e.g. *F*, *t*, *r*) with confidence intervals, effect sizes, degrees of freedom and *P* value noted<br>*Give P values as exact values whenever suitable.* |
| ☒ | ☐ | For Bayesian analysis, information on the choice of priors and Markov chain Monte Carlo settings |
| ☒ | ☐ | For hierarchical and complex designs, identification of the appropriate level for tests and full reporting of outcomes |
| ☒ | ☐ | Estimates of effect sizes (e.g. Cohen's *d*, Pearson's *r*), indicating how they were calculated |

*Our web collection on statistics for biologists contains articles on many of the points above.*

## Software and code

Policy information about availability of computer code

| Data collection | This study is based on the UK Biobank data release. No new data was collected for this study and hence no software was used for data collection. |
|---|---|
| Data analysis | The genotype data was handled with QCTOOL version 2.0.1 (https://www.well.ox.ac.uk/~gav/qctool_v2/) and PLINK version 1.90 (https://www.cog-genomics.org/plink/1.9/) and 2.00 (https://www.cog-genomics.org/plink/2.0/). IBD segments of siblings were inferred with snipar (https://github.com/AlexTISYoung/snipar/blob/ff48c642da1e45067afae1e21f5e5e450d4d4ef9/) and also with KING version 2.2.4 (https://www.kingrelatedness.com) for comparison. Statistical analysis were performed in python version 2.7.11 (https://www.python.org) and R version 3.4.3 (https://www.R-project.org/). LD score regression intercepts and estimates of heritability and genetic correlations were attained with LDSC version 1.0.1 (https://github.com/bulik/ldsc). GWAS summary statistics for the phenotypes shown in Table 1 were attained with BOLT-LMM version 2.3 (https://alkesgroup.broadinstitute.org/BOLT-LMM/). Python and R scripts for performing primary participation GWAS are available at https://github.com/stefaniabe/PrimaryParticipationGWAS. |

For manuscripts utilizing custom algorithms or software that are central to the research but not yet described in published literature, software must be made available to editors and reviewers. We strongly encourage code deposition in a community repository (e.g. GitHub). See the Nature Portfolio guidelines for submitting code & software for further information.

## Data

Policy information about availability of data

All manuscripts must include a data availability statement. This statement should provide the following information, where applicable:
- Accession codes, unique identifiers, or web links for publicly available datasets
- A description of any restrictions on data availability
- For clinical datasets or third party data, please ensure that the statement adheres to our policy

The primary participation GWAS summary statistics generated in this study have been deposited to GWAS catalog under the accession codes GCST90267220, GCST90267221, GCST90267222 and GCST90267223. Researchers can apply for access to individual-level UK Biobank data on their website (http://www.ukbiobank.ac.uk/register-apply/).

# Field-specific reporting

Please select the one below that is the best fit for your research. If you are not sure, read the appropriate sections before making your selection.

☒ Life sciences  ☐ Behavioural & social sciences  ☐ Ecological, evolutionary & environmental sciences

For a reference copy of the document with all sections, see nature.com/documents/nr-reporting-summary-flat.pdf

# Life sciences study design

All studies must disclose on these points even when the disclosure is negative.

| | |
|---|---|
| Sample size | This study is based on all 313,860 genotyped individuals in the UK Biobank that fulfilled our relatedness, ancestry and quality control criteria. The primary participation GWASs were performed with 16,668 sibling pairs and 4,427 parent-offspring pairs. PGS analysis were performed with a non-overlapping group of 272,409 individuals. |
| Data exclusions | The sample was restricted to first-degree relative pairs with white British ancestry (the primary participation GWASs) and white British individuals with no relatives of 3rd degree or closer within the UK Biobank (the PGS analysis). The parent-offspring pairs and sibling pairs were chosen so that the two groups woulds not overlap. For sib-ships with more than two siblings, we chose the two first participating siblings. We excluded all individuals who had withdrawn consent, had a duplicate/twin in sample, had an excess of third degree relatives, were a heterozygosity or missingness outlier, had missing rate above 2%, were not included in the kinship inference of UK Biobank, were not included in the phasing input of UK Biobank, showed potential sex chromosome aneuploidy, demonstrated potential sex mismatch. |
| Replication | For each SNP, association with primary participation was tested in three non-overlapping groups (parent-offspring pairs, IBD1 sibling pairs and IBD2/IBD0 siblings pairs) resulting in three independent test-statistics (TNTC, WSPC and BSPC). |
| Randomization | The methodology introduced in this paper relies on the Mendelian model of inheritance, i.e. one of the two alleles in a parent is randomly transmitted to the offspring with equal probability. |
| Blinding | This is not relevant to our study as we did not compare different experimental groups. |

# Reporting for specific materials, systems and methods

We require information from authors about some types of materials, experimental systems and methods used in many studies. Here, indicate whether each material, system or method listed is relevant to your study. If you are not sure if a list item applies to your research, read the appropriate section before selecting a response.

## Materials & experimental systems

| n/a | Involved in the study |
|---|---|
| ☒ | Antibodies |
| ☒ | Eukaryotic cell lines |
| ☒ | Palaeontology and archaeology |
| ☒ | Animals and other organisms |
| ☐ | ☒ Human research participants |
| ☒ | Clinical data |
| ☒ | Dual use research of concern |

## Methods

| n/a | Involved in the study |
|---|---|
| ☒ | ChIP-seq |
| ☒ | Flow cytometry |
| ☒ | MRI-based neuroimaging |

# Human research participants

| | |
|---|---|
| Population characteristics | The UK Biobank is a prospective cohort that includes approximately 500 thousand genotyped and phenotyped individuals (45.6% male), aged 40-69 years old at recruitment, from across the United Kingdom. |
| Recruitment | Invitations to participate were sent to 9,238,453 individuals who were aged between 40 and 69 years and lived within 25-mile radius of any of the 22 UK Biobank assessment centres. Of those, 5.45% participated and went through baseline assessments that took place from 2006 to 2010. This has been described in detail by Fry et al. (2017) ( https://doi.org/10.1093/aje/kwx246). |
| Ethics oversight | UK Biobank has approval from the North West Multi-centre Research Ethics Committee (MREC) as a Research Tissue Bank (RTB) approval. Research Ethics Committee reference 21/NW/0157. |

Note that full information on the approval of the study protocol must also be provided in the manuscript.

