## [Peer Review File · Nature Genetics]

Peer Review Information

Manuscript Title: Studying the genetics of participation using footprints left on the ascertained genotypes

Corresponding author name(s): Ms Stefania Benonisdottir, Augustine Kong

Reviewer Comments & Decisions:

Decision Letter, initial version:

18th Mar 2022

Dear Ms Benonisdottir,

Your Technical Report, "The Genetics of Participation: Method and Analysis" has now been seen by 3 referees. You will see from their comments copied below that while they find your work of considerable potential interest, they have raised quite substantial concerns that must be addressed. In light of these comments, we cannot accept the manuscript for publication, but would be very interested in considering a revised version that addresses these serious concerns.

In brief, the three referees all acknowledge the importance of the question being addressed. However, they vary in their degree of support for publication; but given there is clear guidance made to address the concerns raised, we believe there is a path to publication.

Reviewer #1 provides a detailed and thoughtful review of your study, with what we think are many useful suggestions. In particular, they think that the simulation studies should be expanded and presented in the main text, and raise an important question regarding the major allele bias observed in your proposed association statistics.

Reviewer #2 is more positive, but thinks that there are still some areas that can be improved that would give further confidence to your method.

Reviewer #3, by contrast, sounds less convinced that there is sufficient advance presented here for publication at Nature Genetics. Nevertheless, they offer useful suggestions that would strengthen the study.

In our reading of these reports, we noted several overlapping suggestions, for example the expansion of the simulations, that would be especially important to address. We think that the major allele bias that is flagged by both Reviewer #1 and #3 is a particularly worrying comment that must be fully

addressed for a successful revision; given that Reviewer #1 has provided detailed suggestions for doing so, we think this should be achievable.

We hope you will find the referees' comments useful as you decide how to proceed. If you wish to submit a substantially revised manuscript, please bear in mind that we will be reluctant to approach the referees again in the absence of major revisions.

To guide the scope of the revisions, the editors discuss the referee reports in detail within the team, including with the chief editor, with a view to identifying key priorities that should be addressed in revision and sometimes overruling referee requests that are deemed beyond the scope of the current study. We hope that you will find the prioritised set of referee points to be useful when revising your study. Please do not hesitate to get in touch if you would like to discuss these issues further.

If you choose to revise your manuscript taking into account all reviewer and editor comments, please highlight all changes in the manuscript text file. At this stage we will need you to upload a copy of the manuscript in MS Word .docx or similar editable format.

*2) If you have not done so already please begin to revise your manuscript so that it conforms to our Technical Report format instructions, available [here](http://www.nature.com/ng/authors/article_types/index.html).

*3) Include a revised version of any required Reporting Summary: <https://www.nature.com/documents/nr-reporting-summary.pdf>

[redacted]

If you wish to submit a suitably revised manuscript we would hope to receive it within 6 months. If you cannot send it within this time, please let us know. We will be happy to consider your revision so long as nothing similar has been accepted for publication at Nature Genetics or published elsewhere. Should your manuscript be substantially delayed without notifying us in advance and your article is eventually published, the received date would be that of the revised, not the original, version.

Thank you for the opportunity to review your work.

Sincerely,

Michael Fletcher, PhD
Associate Editor, Nature Genetics

ORCID: 0000-0003-1589-7087

Referee expertise: all referees are statistical geneticists with experience in GWAS. In addition, Reviewers #2 and #3 have expertise in studies of participation.

Reviewers' Comments:

Reviewer #1:

Remarks to the Author:

This paper investigates the genetic architecture of participation (vs. non-participation) in UK Biobank. The challenge of lack of genetic data from non-participants is addressed by a new method based on genetic segments shared between first-degree relatives who are UK Biobank participants.

The method and empirical findings are potentially interesting. Considerable further work is needed to estimate genetic architecture parameters and validate the robustness of the method via application to other traits.

Major comments:

1. Simulations are essential for evaluating any statistical genetics method. I suggest that the simulations (currently presented in the Supp. Text and Extended Data Figure 9) should be expanded and presented in a Simulations subsection of Results, citing at least one main Figure and appearing

before the analysis of participation in UK Biobank. Details of the simulation framework should be provided in the Methods section.

The simulations should contain a number of first-degree relative pairs that is similar to the analysis of participation in UK Biobank, in order to assess power.

The simulations should check that there is a 100% genetic correlation (Bulik-Sullivan et al. 2015b Nat Genet; Wu et al. 2022 Am J Hum Genet) between the DPO/DSIB1/DSIB20 association statistics (from cases) and case-control association statistics.

The simulations should also compare the power of these respective statistics (e.g. by comparing average χ^2 at causal SNPs, which are known within these simulations), and check DPO/DSIB1/DSIB20 association statistics for correct calibration at null SNPs (which are known within these simulations).

2. I suggest that the paper should provide estimates of the SNP-heritability (liability scale or observed scale) and prevalence of participation in UK Biobank, or at least constrain the possible values of these parameters. Note that for case-control association statistics there is a simple relationship between average χ^2 and observed-scale SNP-heritability (Yang et al. 2011 Eur J Hum Genet). The simulations may be valuable in extending this relationship to DPO/DSIB1/DSIB20 association statistics.

3. I suggest that the paper should apply the DPO/DSIB1/DSIB20 method to binary traits within UK Biobank (e.g. a binary trait defined by top 50% or top 10% of height (which has the advantage of being highly heritable), or a high-prevalence disease trait) and compare the resulting association statistics to standard case-control association statistics—which can be computed for UK Biobank binary traits. My biggest concern is that the DPO/DSIB1/DSIB20 association statistics might include a combination of artifactual and true signals (given that there is a major allele artifact that is corrected post hoc, how can we be sure that there are no other artifacts?). This analysis is critical to addressing that concern.

The paper should check that this results in a 100% genetic correlation (Bulik-Sullivan et al. 2015b Nat Genet; Wu et al. 2022 Am J Hum Genet) between the DPO/DSIB1/DSIB20 association statistics (from cases) and case-control association statistics, and that genome-wide significant SNPs in the DPO/DSIB1/DSIB20 association statistics (without applying a genomic control correction) replicate in the case-control analysis.

In addition, the paper should check that this results in realistic values of average χ^2 of DPO/DSIB1/DSIB20 association statistics and resulting estimates of SNP-heritability.

For traits known to be impacted by indirect effects (e.g. college education; Kong et al. 2018 Science, ref. 9) some divergence between DPO/DSIB1/DSIB20 association statistics and case-control association statistics may be expected, because DPO/DSIB1/DSIB20 association statistics exclude indirect effects. It would be interesting for the paper to check this, e.g. including college education in the set of binary traits within the UK Biobank to which the DPO/DSIB1/DSIB20 method is applied.

4. I suggest that the paper should estimate and report genetic correlations (Bulik-Sullivan et al. 2015b Nat Genet; Wu et al. 2022 Am J Hum Genet) between participation in UK Biobank and the traits listed in Table 1. The PGS correlations that are listed are related to genetic correlations, but are less

interpretable (for example, PGS correlations are impacted by sample size, and are thus expected to be <100% even for identical traits).

The genetic correlations could be estimated either by applying a method that estimates genetic correlations (Bulik-Sullivan et al. 2015b Nat Genet; Wu et al. 2022 Am J Hum Genet), or by transforming PGS correlations into genetic correlations (Dudbridge 2013 PLoS Genet).

Minor comments:

p.1 (Title): I find the words "method" and "analysis" to be rather boring words to include in the title of a paper. It would be better to include words that characterize the method and/or empirical findings.

p.1 (Abstract): Effect sizes (e.g. genetic correlations; see Major comment 4) would be more interesting to report in the Abstract than P-values, which are impacted by sample size and not informative for effect sizes.

p.2: "further engagement in optional components of the study has been demonstrated to have associations with both genotypes and phenotypes [ref. 4-7]": please provide more details here on those published findings, which constitute critical background.

p.3-7: The content defining DPO, DSIB1, DSIB20 and citing Figure 1 does not belong in the Introduction. This content should be included in an initial Methods overview subsection of Results.

p.3-7: This content is too detailed for a Methods overview subsection of Results. Most of it should be moved to a Methods section that appears after the Discussion section. (The main text does not currently contain a Methods section, but it should.) All that is needed in the Methods overview subsection of Results is a definition of DPO, DSIB1, DSIB20 using text and a statement that these statistics employ appropriate matching to avoid indirect effects or population stratification, citing the Methods section for further details.

p.3-7: I suggest that 0 equations is optimal for the Methods overview subsection of Results, and 1 equation should be a maximum. The Methods section can contain 5 (or more) equations.

Equations 1-5: if I understand correctly, DPO, DSIB1, DSIB20 (and the values of nPO, nSIB1, nSIB2, nSIB0) are specific to each SNP. I suggest that the paper should either explicitly index each of these variables using a SNP index, or very clearly state that each of these quantities is implicitly indexed by SNP and computed separately for each SNP in turn.

Equations 1-5: I suggest that every equation in the paper (and Supp. Text) should be numbered.

Figure 1: I find panel A to be misleading, because not all first-degree relative pairs have IBD=1 at a locus (as is clear from subsequent panels). I suggest to delete panel A.

p.5: "The case where both parents are ascertained together with an offspring can be treated as two parent-offspring pairs as the transmission from the two parents are independent under the null hypothesis": I agree with this statement. However, I suggest that the paper should explicitly state whether or not there are any other instances where it is permissible for a UK Biobank participant to be part of two distinct first-degree relative pairs used to compute association statistics for UK Biobank participation. If yes, then the paper must explain why this does not cause association statistics in null

data (specifically, the combination of DPO, DSIB1, DSIB20) to deviate from the null distribution.

p.7: “they can have different expectations depending on the nature of the ascertainment bias”: the paper is deficient in not exploring in more detail whether or not the three statistics produce discordant results in analyses of participation in UK Biobank (e.g. in the pairwise genetic correlation between them, or in the respective genetic correlations with traits listed in Table 1) and drawing resulting inferences about the nature of the ascertainment bias. This content must be provided.

p.7-10: For the section titled “UK Biobank and Data Processing Artefacts” (citing Figure 2), I suggest that most of this content should be moved to a Methods section that appears after the Discussion section. (The main text does not currently contain a Methods section, but it should.) A summary of this content (citing Figure 2) should be included in the Methods overview subsection of Results.

p.9: “After the above adjustments, the tendency for the major allele to be positively associated with participation was substantially reduced but not completely eliminated”: the paper is deficient in not providing further details of this bias, and must cite a Supplementary Table thoroughly quantifying this bias, both before and after the adjustments described. This can be cited either from the Methods overview subsection of Results, or from the Methods section. (It should also be cited on p.25 of the Supplementary material, which is similarly deficient in not providing further details of this bias.)

p.10: “IBD sharing status (0, 1 or 2) of every SNP was ‘called’ using the program KING”: the paper should clarify whether phased or unphased data was used as input to KING.

p.10: “IBD sharing status (0, 1 or 2) of every SNP was ‘called’ using the program KING”: the paper should note and cite the development and application of more recent IBD inference algorithms to UK Biobank data (e.g. Nait Saada et al. 2020 Nat Commun) and justify the decision to instead use KING (Manichaikul et al. 2010 Bioinformatics). Would using a cutting-edge IBD inference algorithm avoid some of the biases observed, such as the tendency for the major allele to be positively associated with participation?

p.11: I suggest that the first 3 lines of this page, which also pertain to methodology for computing association statistics, should be moved to a Methods overview subsection of the Results section.

p.11: Genomic control: it is widely accepted that genomic inflation factors are expected for polygenic traits (Yang et al. 2011 Eur J Hum Genet). The paper should report the genomic inflation factor (of association statistics for UK Biobank participation) prior to applying genomic control. If the genomic inflation factor is substantial, this would be a major concern—particularly since the statistics are designed to be immune to population stratification—and would require considerable caveats.

p.11: “we derived the weights of a participation polygenic scores [sic] (pPGS) based on our participation GWAS (Supp. Text)”: the method used to compute the pPGS is central to this paper, and should be provided in the main text (Results section or Methods section).

p.11: “we derived the weights of a participation polygenic scores [sic] (pPGS) based on our participation GWAS (Supp. Text)”: looking at the Supp. Text, it seems as if raw allele counts were weighted by standardized T-statistics? However, it would be more appropriate to either weight raw allele counts by estimated per-allele effect sizes, or weight standardized allele counts (standardized to mean 0 and variance 1, for each SNP) by standardized T-statistics. Please clarify and justify the choice that was made, and change if necessary.

p.11: "we derived the weights of a participation polygenic scores [sic] (pPGS) based on our participation GWAS (Supp. Text)": looking at the Supp. Text, it is a bit odd that the pPGS was constructed by simply summing together effects of all SNPs passing QC, given that both (i) crude methods based on P-value thresholding (Purcell et al. 2009 Nature) and (ii) more sophisticated Bayesian methods (Lloyd-Jones et al. 2019 Nat Commun) have proven far superior in practice. Perhaps only a bit odd because I hypothesize that this might not make a big difference for highly polygenic traits (which is likely the case for participation), however, this paper must either explore more sophisticated PGS methods or state that the failure to explore more sophisticated PGS methods is a limitation of the paper.

p.11: "consistent with the known differences in EA and BMI between sample and population": I agree that the signs of the PGS correlations are consistent with the signs of the known differences. I suggest that the paper should discuss whether it is further possible, using inferred prevalence (see Major comment 2) and genetic correlations (see Major comment 4), to assess whether the magnitudes of genetic correlations are consistent with the magnitudes of known differences in EA and BMI between sample and population.

p.15: "while it has been reported that UKBB participation rates differ by sex and age [ref. 17], the pPGS is associated with neither ($P > 0.05$). This implies that the effect of the pPGS, based on autosomal variants, is additive to the effects of sex and age on participation, with no detectable statistical interactions." Although this statement might be technically correct, I find it to be misleading. In light of published evidence of "differential participation effects of the sexes [ref. 8]", it is expected that the genetic architecture of male participation and the genetic architecture of female participation would be different. The current analysis, which uses a pPGS based on male+female participation, does not address this question. The proper way to address this question would be to apply the DPO/DSIB1/DSIB20 method to male-male pairs and female-female pairs separately, and assess differences (e.g. genetic correlation < 1) between the genetic architecture of male participation and the genetic architecture of female participation. Either this analysis should be performed, or it should be made clear (in the Results and/or Discussion section) that this is the proper analysis but it has not been performed.

p.15: overrepresentation of close relatives: "Here we show that shared genetics most likely also play a substantial role": I find this text to be overstated. Please either provide a quantification of the contribution of shared genetics to overrepresentation of close relatives, or change "substantial" to "greater than zero".

General comment: I suggest to define and use a single term for the trait being studied. The paper currently uses "participation", "ascertained", "ascertainment probability" interchangeably, but it would be better to use only one of these terms. I suggest "participation" or "study participation". I do not favor "ascertained" or "ascertainment probability", as ascertainment is a term that has many different meanings and uses.

General comment: Perhaps give the method a name? "DPO/DSIB1/DSIB20 method" (see above) may not be the best name.

General comment: this is not the first disease/trait analysis method to make use of shared genetic segments in related samples. Would it be of value to cite literature on linkage mapping (e.g. Ott et al. 2015 Nat Rev Genet) and population-based IBD mapping (e.g. Browning et al. 2012 Genetics)?

Reviewer #2:

Remarks to the Author:

This is a very important paper which addresses a very contemporary point in the discussion around GWAS studies. In particular the authors have developed a new methods for deriving a GWAS of participation without having measured non-participants but exploiting the IBD regions from sib pairs. The approach is very clever and of clear interest and usefulness.

Although overall the work is fine and well conducted the paper is a bit difficult to follow especially in the methods. Clearly the authors have everything very clear in their heads but most readers will not. I think the paper would benefit for a bit more clarity in the explanation.

This said I do have a few comments for the authors:

- 1) The authors should mention the explanation of why the IBD segments will be enriched in participation alleles compared to the non-shared ones. I went through the whole paper and found no mention of the simulations which are in the supplement apart from the legend of figure 9. The authors should point this out in the main text otherwise their theorem seems postulated.
- 2) Following up on the previous comment, the demonstration of the enrichment of participation alleles in IBD segments is extremely hard to follow as often new parameters are introduced without stating what they are. The reader has to deduce it from the rest of the text but it makes reading it very hard.
- 3) What does DPO stand for? I don't seem to be able to find any extended definition.
- 4) There is a similar problem with DSIB1 which is never defined anywhere.
- 5) It would help if formulas were numbered.
- 6) I think something that is missing from the paper is an actual comparison between the method and running a GWAS of participation. I realise that just selecting out random samples from the UK biobank may not accomplish the task given that the authors postulate that the differences are embedded in the participants allelic frequencies, but at least simulations could be performed.
- 7) The study makes another strong assumption which is that the participating sib pairs or parent-offspring pairs are representative of the rest of UK biobank. Is this true? Although the authors say their results are unbiased from population stratification, this doesn't say anything about generalisability. It could be that these participants are more biased than the rest of UK biobank. This could be tested directly through GWAS of people with relatives vs the rest.
- 8) Looking at the frequency correction, it is unclear to me why the authors include only odd powers in the formula and why up to the power of 5.
- 9) Why were quantitative traits standardised separately? This may induce bias in the results as they will refer to 1SD which corresponds to a different number in the two sexes. For example for BMI women have a higher variance than men and thus dividing the values by these numbers will be effectively moving the data from the same scale (BMI) to two distinct ones which may influence the results.

10) The authors have used only 40 PCs to correct for population stratification. Previous studies have shown that this is not enough (see for example PMID: 31636407). They should either use more or use Grammar-Gamma residuals as implemented in GCTA for example.

Reviewer #3:

Remarks to the Author:

1. The subject of this paper is interesting and important – that participation in genetic or other studies is a heritable trait with many phenotypic and genetic associations. The abstract slightly overstates some of the background- that it is not possible to examine participation directly (it is, studies such as UK Biobank have looked at later optional components) and these studies (as well as studies looking at the phenotypic associations of participation) have made similar discoveries.

2. The abstract contains several results with p-values in parenthesis after the text, but no effect size. This is not best practice – there should be effect sizes (R^2 , or standardised betas) accompanying each p-value, to aid interpretation. The polygenic score is for participation (participation implicitly coded 1, non-participation = 0) – but that is not clear in the abstract.

3. The examples given (presidential candidate voting, vaccination status) are vague and it isn't made clear in the next sentence what hypothesis is being tested in these 'fictional studies' and what is hypothetically being adjusted for in the next.

4. I understand the genetic principles set out in the introduction, that alleles associated with participation will be more common in segments that are present in multiple participants. I do not follow why this would necessarily be immune to population stratification. Furthermore, alleles correlated between segments on different chromosomes of the same individual are known to be correlated when there is assortative mating in a population. This has been demonstrated for educational attainment, with which they find a strong association. How can we be confident that the results shown do not represent the effects of assortative mating within a population?

5. I assume from following the definition of the statistics that the F terms are the allele frequencies in the shared/non-shared segments (FPT = frequency of the 1-coded allele in segments transmitted from parents to offspring; FPNT = frequency of the 1-coded allele in segments not transmitted from parents and offspring). Could this be pointed out more explicitly in the text as an aid to readers?

6. Is the derivation of the standard error for the DPO, DSIB1, and DS1B20 statistics (referred to on Line 206) given anywhere?

7. The major allele bias seems largely unexplained and, without a reasonable explanation. This raises the general possibility that, in the absence of a strong/proven explanation for this and the multiple 'spurious' MHC variants – other sources of bias may be lurking? For example, would a small proportion of minor allele homozygotes being miscalled as heterozygotes lead to such bias? The authors also state that "part of this 'major allele effect' might be real" (Lines 184-185). Can they explain what real processes would lead to this effect? This paper would perhaps be better suited to a more technical journal where these issues can be addressed.

8. Why is it necessary to GC-adjust? This is a now infrequently used approach in GWAS, leading to

over-correction when the signal is polygenic. What is the justification for using it in the current circumstances?

9. It's not clear how the authors chose to present some traits that were not associated with pPGS and not others. Why not apply a PheWAS approach and provide a plot that allows the reader to look at phenotype domains and how each might be impacted by participation bias? Similarly, UKBB has other optional assessments (aide-memoire form, Imaging visit, Mental Health Questionnaire, Healthy Work questionnaire, Cognitive function online follow-up) in addition to the diet-by-recall and the physical activity monitoring that were reported in the study. Were these two follow-up assessments analysed for a specific reason or could the analysis be expanded to include the other assessments? Being more comprehensive would make the present study easier to compare to previous studies on the genetics of optional participation in UKBB.

10. Minor issues: This sounds awkward: 'identifying such (a) component' [abstract]. Code available on publication – but not now, so that reviewers can spot errors?

Author Rebuttal to Initial comments

Point by Point Responses to Reviewers

Reviewer 1

This paper investigates the genetic architecture of participation (vs. non-participation) in UK Biobank. The challenge of lack of genetic data from non-participants is addressed by a new method based on genetic segments shared between first-degree relatives who are UK Biobank participants.

The method and empirical findings are potentially interesting. Considerable further work is needed to estimate genetic architecture parameters and validate the robustness of the method via application to other traits.

We thank the reviewer for the valuable comments. We recommend the reviewer to start by reading the General Responses above, as they address many of the key concerns of the reviewer, e.g. the nature of the major allele bias, the different behavior of the three test statistics, genetic correlations between primary participation and other traits, as well as its SNP-heritability.

Major Comments

(1) Simulations are essential for evaluating any statistical genetics method.

I suggest that the simulations (currently presented in the Supp. Text and Extended Data Figure 9) should be expanded and presented in a Simulations subsection of Results, citing at least one main Figure and appearing before the analysis of participation in UK Biobank.

Details of the simulation framework should be provided in the Methods section.

The simulations should contain a number of first-degree relative pairs that is similar to the analysis of participation in UK Biobank, in order to assess power.

The simulations should check that there is a 100% genetic correlation (Bulik-Sullivan et al. 2015b Nat Genet; Wu et al. 2022 Am J Hum Genet) between the DPO/DSIB1/DSIB20 association statistics (from cases) and case-control association statistics.

The simulations should also compare the power of these respective statistics (e.g. by comparing average χ^2 at causal SNPs, which are known within these simulations), and check DPO/DSIB1/DSIB20 association statistics for correct calibration at null SNPs (which are known within these simulations).

Thanks for the comment. We have performed many simulations to understand (i) the sampling properties of our methods and (ii) the nature of the major allele bias. With (i), ignoring data artefacts, we performed simulations that show the relationship between the difference in allele frequency between the population and the sample and the difference in allele frequency between the shared and not shared alleles (Figure 4). We also show that the WSPC (DSIB1) and the BSPC (DSIB20) are capturing practically the same effects mainly because they are based on the same set of sib-pairs. These results allow us to calculate the effective sample size of our comparisons relative to a standard case-control analysis (described in Methods), which is needed for us to

calculate the estimated heritability of the underlying liability score. With (ii), we have performed simulations showing how genotyping errors, even when completely random, would lead to a major allele bias for WSPC and TNTC (DPO). Together with our analysis on the impact of phasing errors, we show that genotyping errors are most likely responsible for a substantial part of the major allele bias for variants with low minor allele frequency (MAF). As suggested by the reviewer, the simulation results (i) are discussed in the main text and highlighted in Figure 4.

(2) I suggest that the paper should provide estimates of the SNP-heritability (liability scale or observed scale) and prevalence of participation in UK Biobank, or at least constrain the possible values of these parameters. Note that for case-control association statistics there is a simple relationship between average χ^2 and observed-scale SNP-heritability (Yang et al. 2011 Eur J Hum Genet). The simulations may be valuable in extending this relationship to DPO/DSIB1/DSIB20 association statistics.

As part of the calculations performed using LD score regression to estimate genetic correlations, using the BSPC results, we have an estimate of 12.5% for the liability score underlying primary participation. This is now reported in the main text. The technical details of those computations are described in Methods.

(3) I suggest that the paper should apply the DPO/DSIB1/DSIB20 method to binary traits within UK Biobank (e.g. a binary trait defined by top 50% or top 10% of height (which has the advantage of being highly heritable), or a high-prevalence disease trait) and compare the resulting association statistics to standard case-control association statistics—which can be computed for UK Biobank binary traits. My biggest concern is that the DPO/DSIB1/DSIB20 association statistics might include a combination of artifactual and true signals (given that

there is a major allele artifact that is corrected post hoc, how can we be sure that there are no other artifacts?). This analysis is critical to addressing that concern.

The paper should check that this results in a 100% genetic correlation (Bulik-Sullivan et al. 2015b Nat Genet; Wu et al. 2022 Am J Hum Genet) between the DPO/DSIB1/DSIB20 association statistics (from cases) and case-control association statistics, and that genome-wide significant SNPs in the DPO/DSIB1/DSIB20 association statistics (without applying a genomic control correction) replicate in the case-control analysis.

In addition, the paper should check that this results in realistic values of average χ^2 of DPO/DSIB1/DSIB20 association statistics and resulting estimates of SNP-heritability.

For traits known to be impacted by indirect effects (e.g. college education; Kong et al. 2018 Science, ref. 9) some divergence between DPO/DSIB1/DSIB20 association statistics and case-control association statistics may be expected, because DPO/DSIB1/DSIB20 association statistics exclude indirect effects. It would be interesting for the paper to check this, e.g. including college education in the set of binary traits within the UK Biobank to which the DPO/DSIB1/DSIB20 method is applied.

We agree with the reviewer's concerns that artificial signals are mixed in with the true signals, as reflected by the major allele bias in the TNTC and WSPC results. Most importantly, with our updated analyses for the revised manuscript, we believe that the results from BSCP are now essentially capturing true signals only. The TNTC and WSPC are nonetheless capturing true signals together with some artificial ones. The latter results from having to split genotypes into shared and not-shared alleles, which is sensitive to both simple genotyping errors and phasing errors. More details on the nature of the bias and the adjustments we performed to address the

problem are in the General Responses. Overall, we believe our updated results provide a good understanding of the power of the three test statistics in capturing the true signals and the problems caused by of data errors. The number of sib-pairs in the UKBB, for our purpose, is not that large compared to sample sizes in traditional GWASs. By selecting participants based on their height or EA values will further reduce the sample size of first-degree relatives. Most importantly, given our updated results and improved understanding, it is not clear to us that such approach would further increase our understanding of true and artificial signals.

(4) I suggest that the paper should estimate and report genetic correlations (Bulik-Sullivan et al. 2015b Nat Genet; Wu et al. 2022 Am J Hum Genet) between participation in UK Biobank and the traits listed in Table 1. The PGS correlations that are listed are related to genetic correlations, but are less interpretable (for example, PGS correlations are impacted by sample size, and are thus expected to be <100% even for identical traits). The genetic correlations could be estimated either by applying a method that estimates genetic correlations (Bulik-Sullivan et al. 2015b Nat Genet; Wu et al. 2022 Am J Hum Genet), or by transforming PGS correlations into genetic correlations (Dudbridge 2013 PLoS Genet).

We have now estimated genetic correlations between primary participation and the traits listed in Table 1 using the BSPC (DSIB20) results. The significant results, i.e. 95% CI does not include 0, are presented in the current Table 2. Estimates based on the adjusted WSPC (DSIB1) results are given in Extended Data Table 2. We agree with the reviewer that these results add valuable insight for the understanding of participation as a genetic trait.

Minor comments:

(5) p.1 (Title): I find the words “method” and “analysis” to be rather boring words to include in

the title of a paper. It would be better to include words that characterize the method and/or empirical findings.

Given that the manuscript is submitted as a Technical Report, we thought a ‘boring’ title would be fine. A more catchy title we also considered is: “Studying the Genetics of Participation using Samples only”. This highlights what is the most special aspect of the manuscript. If the editor found this alternative appealing, we would be happy to change the title.

(6) p.1 (Abstract): Effect sizes (e.g. genetic correlations; see Major comment 4) would be more interesting to report in the Abstract than P-values, which are impacted by sample size and not informative for effect sizes.

We highlight *P*-values in the abstract because the main point we want to make is that the participation genome-scan, which is performed with samples only and without phenotypes, can nonetheless capture genetic signals on participation. We do agree with major comment 4 that genetic correlations are important, which we now do have in Table 2. We have also noted in the abstract, that the estimated genetic correlation between primary participation and educational attainment is 36.7%.

(7) p.2: “further engagement in optional components of the study has been demonstrated to have associations with both genotypes and phenotypes [ref. 4-7]”: please provide more details here on those published findings, which constitute critical background.

We have now added the following information about those published findings to the main text on p. 2 (see in bold below):

For genetic investigations, among participants of the primary study who have contributed DNA, further engagement in optional components of the study, **such as filling out additional questionnaires, showing up for follow-up assessments or providing a valid email address**, has been demonstrated to have associations with both genotypes and phenotypes

(8) p.3-7: The content defining DPO, DSIB1, DSIB20 and citing Figure 1 does not belong in the Introduction. This content should be included in an initial Methods overview subsection of Results.

Following the reviewer's comment, we have simplified the description of DPO, DSIB1 and DSIB20 in the main text (now called TNTC, WSPC and BSPC) and have added a Method section after the Discussion section in which we give detailed descriptions about the t-statistics computed from the three frequency differences.

(9) p.3-7: This content is too detailed for a Methods overview subsection of Results. Most of it should be moved to a Methods section that appears after the Discussion section. (The main text does not currently contain a Methods section, but it should.) All that is needed in the Methods overview subsection of Results is a definition of DPO, DSIB1, DSIB20 using text and a statement that these statistics employ appropriate matching to avoid indirect effects or population stratification, citing the Methods section for further details.

p.3-7: I suggest that 0 equations is optimal for the Methods overview subsection of Results, and 1 equation should be a maximum. The Methods section can contain 5 (or more) equations.

Thank for the suggestions. We have moved most details into Methods. However, given that this is a Technical Report, and the comparisons are arguably its most novel aspect, we believe that having them clearly defined in the main text is justified.

(10) Equations 1-5: if I understand correctly, DPO, DSIB1, DSIB20 (and the values of nPO, nSIB1, nSIB2, nSIB0) are specific to each SNP. I suggest that the paper should either explicitly index each of these variables using a SNP index, or very clearly state that each of these quantities is implicitly indexed by SNP and computed separately for each SNP in turn.

We thank the reviewer for pointing this out. We have now added the following text on pages 4 and 6 (see in bold below) to make it clear that the frequency statistics are specific to each SNP:

‘For an ascertained parent-offspring pair where the other parent is not ascertained, there are three distinct alleles by descent **for a given SNP.**’

‘For each sib-pair that has IBD state 1 **for a given SNP** (Fig. 1C), there are one distinct share allele (S) and two distinct not-shared alleles (NS_1 and NS_2).’

(11) Equations 1-5: I suggest that every equation in the paper (and Supp. Text) should be numbered.

Thanks for the suggestion. We now have the equations numbered.

(12) Figure 1: I find panel A to be misleading, because not all first-degree relative pairs have IBD=1 at a locus (as is clear from subsequent panels). I suggest to delete panel A.

Panel A is for illustrating the shared versus not-shared comparison for ‘all’ (not just first-degree) relative pairs that happen to share one allele IBD at a locus. It does not suggest all first-degree relative pairs have IBD=1 at a locus. It is meant to illustrate, ‘when’ a relative pair is IBD1 at a

locus, the shared versus not shared comparison can be made. We have added the following text to the figure legend to clarify this (see in bold below):

a) General relative pairs **sharing a locus IBD=1**.

(13) p.5: “The case where both parents are ascertained together with an offspring can be treated as two parent-offspring pairs as the transmission from the two parents are independent under the null hypothesis”: I agree with this statement. However, I suggest that the paper should explicitly state whether or not there are any other instances where it is permissible for a UK Biobank participant to be part of two distinct first-degree relative pairs used to compute association statistics for UK Biobank participation. If yes, then the paper must explain why this does not cause association statistics in null data (specifically, the combination of DPO, DSIB1, DSIB20) to deviate from the null distribution.

In this manuscript, we are introducing a way to use information that has not been previously utilized. Our primary goal is to demonstrate that there are valid ways of using this information/ data to investigate the genetics of participation. There most likely is extra information in the data that can also be utilized. However, doing so properly would require extra thought and extra care. One example we point out in the manuscript is including more distant relative pairs in the analysis. We feel that discussing all possible ways of getting more information out of the data in a valid manner is beyond the scope of the current manuscript.

(14) p.7: “they can have different expectations depending on the nature of the ascertainment bias”: the paper is deficient in not exploring in more detail whether or not the three statistics produce discordant results in analyses of participation in UK Biobank (e.g. in the pairwise genetic correlation between them, or in the respective genetic correlations with traits listed in

Table 1) and drawing resulting inferences about the nature of the ascertainment bias. This content must be provided.

We have addressed this in the revision. Empirically, in the updated Table 1, results for PGSs constructed from the three comparisons separately are given, showing that the WSPC and BSPC PGSs have comparable predictive power. The TNTC PGS has lower predictive power which we believe is mostly due to having a lower sample size. In simulations (Figure 4), with respect to capturing real participation signals, under an additive model and for variants which individually have small effects, WSPC and BSPC, as they are utilizing the same set of sib-pairs, should have practically the same (true) effect size. There is some minor difference under a dominant or recessive model, which we also describe in the main text in the revised manuscript. With true effects, the TNTC analyses, which use relative pairs that are not symmetrical, do not need to have the same effect sizes, even after taking sample size into account. However, results in Table 1 suggest they are not too different. With regard to ‘signals’ that result from data and data processing errors, we believe that the BSPC results in the revised manuscript are essentially not affected by artefacts. By comparison, TNTC and WSPC are affected by genotyping errors and phasing errors through the step where a genotype is split into the shared and not shared alleles. Adjustments and consequences are described in detail in the main text, Methods and Supplementary Note Section 2.

(15) p.7-10: For the section titled “UK Biobank and Data Processing Artefacts” (citing Figure 2), I suggest that most of this content should be moved to a Methods section that appears after the Discussion section. (The main text does not currently contain a Methods section, but it should.) A summary of this content (citing Figure 2) should be included in the Methods overview subsection of Results.

We thank the reviewer for this suggestion. We have now added a Method section after the Discussion section in which we give detailed description about the artefacts and how we adjust for them.

(16) p.9: “After the above adjustments, the tendency for the major allele to be positively associated with participation was substantially reduced but not completely eliminated”: the paper is deficient in not providing further details of this bias, and must cite a Supplementary Table thoroughly quantifying this bias, both before and after the adjustments described. This can be cited either from the Methods overview subsection of Results, or from the Methods section. (It should also be cited on p.25 of the Supplementary material, which is similarly deficient in not providing further details of this bias.)

As noted in the general responses, our updated results have provided us with a much better understanding of the causes and consequences of the major allele bias; they are due to data problems as opposed to sampling issues. These are described in the revised main text and with more details in the Methods section. Most importantly, this bias is not an issue for the BSPC analysis as it does not require splitting genotypes into shared and not-shared alleles.

(17) p.10: “IBD sharing status (0, 1 or 2) of every SNP was ‘called’ using the program KING”: the paper should clarify whether phased or unphased data was used as input to KING.

See response to next comment.

(18) p.10: “IBD sharing status (0, 1 or 2) of every SNP was ‘called’ using the program KING”: the paper should note and cite the development and application of more recent IBD inference algorithms to UK Biobank data (e.g. Nait Saada et al. 2020 Nat Commun) and justify the

decision to instead use KING (Manichaikul et al. 2010 Bioinformatics). Would using a cutting-edge IBD inference algorithm avoid some of the biases observed, such as the tendency for the major allele to be positively associated with participation?

We now use the IBD estimation program *snipar* that is part of a recent publication (Young et al., *Nature Genetics* **54**, 897-9052022). It performs much better than *KING*, probably due to the fact that *snipar* is specially designed for sib-pairs, while *KING* isn't. As noted in Methods in the revised manuscript, by using *snipar* and trimming of 250 SNPs at the edges of each IBD segment, we believe the impact of IBD-estimation errors to be negligible. Genotypes of the 'phased' data from UKBB are used as input to *snipar*, but the phasing information is not utilized by *snipar*. We have added the following sentence to the Methods section on p.27 to explain this:

'Neither program takes phasing information into account.'

(19) p.11: I suggest that the first 3 lines of this page, which also pertain to methodology for computing association statistics, should be moved to a Methods overview subsection of the Results section.

We thank the reviewer for this suggestion. We have now moved those three lines to the Method section and the beginning of the Results section (p. 11) now reads as follows:

'We tested 500,632 biallelic sequence variants for association with primary participation in the UKBB.'

(20) p.11: Genomic control: it is widely accepted that genomic inflation factors are expected for polygenic traits (Yang et al. 2011 Eur J Hum Genet). The paper should report the genomic

inflation factor (of association statistics for UK Biobank participation) prior to applying genomic control. If the genomic inflation factor is substantial, this would be a major concern—particularly since the statistics are designed to be immune to population stratification—and would require considerable caveats.

This is now addressed. In particular, no genomic control is needed for the BSPC results because, as noted before, the fitted intercept when applying LDSC to the χ^2 statistics is practically 1. For the TNTC and WSPC, we performed a version of genomic control that is minor-allele-frequency (MAF) dependent as described in Methods. For WSPC, the adjustment factor ranges from 1.02 for MAF = 0.50 to 1.13 to MAF = 0.01. This highlights that genotyping and phasing errors create a substantially higher inflation of χ^2 statistics for the low frequency variants than the common variants. This is now mentioned in the main text (p. 11) and described in detail in Methods and demonstrated in Extended Data Figure 8.

(21) p.11: “we derived the weights of a participation polygenic scores [sic] (pPGS) based on our participation GWAS (Supp. Text)”: the method used to compute the pPGS is central to this paper, and should be provided in the main text (Results section or Methods section).

We have now added a short section in Methods (p. 39) about how the pPGS were computed.

(22) p.11: “we derived the weights of a participation polygenic scores [sic] (pPGS) based on our participation GWAS (Supp. Text)”: looking at the Supp. Text, it seems as if raw allele counts were weighted by standardized T-statistics? However, it would be more appropriate to either weight raw allele counts by estimated per-allele effect sizes, or weight standardized allele counts (standardized to mean 0 and variance 1, for each SNP) by standardized T-statistics. Please clarify and justify the choice that was made, and change if necessary.

See response to the comment below.

(23) p.11: “we derived the weights of a participation polygenic scores [sic] (pPGS) based on our participation GWAS (Supp. Text)”: looking at the Supp. Text, it is a bit odd that the pPGS was constructed by simply summing together effects of all SNPs passing QC, given that both (i) crude methods based on P-value thresholding (Purcell et al. 2009 Nature) and (ii) more sophisticated Bayesian methods (Lloyd-Jones et al. 2019 Nat Commun) have proven far superior in practice. Perhaps only a bit odd because I hypothesize that this might not make a big difference for highly polygenic traits (which is likely the case for participation), however, this paper must either explore more sophisticated PGS methods or state that the failure to explore more sophisticated PGS methods is a limitation of the paper.

We have explored different ways of constructing PGS, and found the current ‘simple’ approach results in a PGS that gives results that are consistent with PGS constructed with nominal weights as well as more ‘sophisticated’ approaches such as LDpred and LD clumping. Furthermore, for traits like EA, the simple PGS explains more of the trait variance. Optimal way of constructing PGS is not the focus of this manuscript. The main point of the PGS analyses is to show that our approach that uses previously ‘unnoticed’ information can capture genetic effects to participation using sample only. That is what we considered to be the major breakthrough/novelty. Finally, an important part of the revised manuscript is the comparison of the PGSs constructed with BSPPC, WSPC and TNTC (see Table 1). For this purpose, we feel a simple PGS is preferred as more complicated approaches would make such comparison less clear as the PGSs might differ because of some details in the more complicated PGS approach. By using the simple approach, the only difference between BSPPC, WSPC and the TNTC PGS are the weights from our primary participation GWASs. The GWAS summary statistics will be made available upon publication

which will enable other researchers to explore more ways of computing primary participation PGSs.

(24) p.11: “consistent with the known differences in EA and BMI between sample and population”: I agree that the signs of the PGS correlations are consistent with the signs of the known differences. I suggest that the paper should discuss whether it is further possible, using inferred prevalence (see Major comment 2) and genetic correlations (see Major comment 4), to assess whether the magnitudes of genetic correlations are consistent with the magnitudes of known differences in EA and BMI between sample and population.

In the current Table 2, we have estimated genetic correlations with EA (0.367) and BMI (-0.220). Trying to understand the multivariate-correlation between these traits and their genetic components, which probably also involve both direct and indirect genetic effects, is, we feel, beyond the scope of the current paper. We do agree that this path of investigations is worthwhile, and we hope the method and results presented in the current manuscript would be an aid to that effort.

(25) p.15: “while it has been reported that UKBB participation rates differ by sex and age [ref. 17], the pPGS is associated with neither ($P > 0.05$). This implies that the effect of the pPGS, based on autosomal variants, is additive to the effects of sex and age on participation, with no detectable statistical interactions.” Although this statement might be technically correct, I find it to be misleading. In light of published evidence of “differential participation effects of the sexes [ref. 8]”, it is expected that the genetic architecture of male participation and the genetic architecture of female participation would be different. The current analysis, which uses a pPGS based on male+female participation, does not address this question. The proper way to address this question would be to apply the DPO/DSIB1/DSIB20 method to male-male pairs and female-

female pairs separately, and assess differences (e.g. genetic correlation < 1) between the genetic architecture

of male participation and the genetic architecture of female participation. Either this analysis should be performed, or it should be made clear (in the Results and/or Discussion section) that this is the proper analysis but it has not been performed.

We have added a sentence on page 17 stating that the analyses we performed were designed to capture the genetic component to participation that is similar for the the two sexes, as opposed to the component that is very different between the sexes. In particular, we state that the fact that our polygenic scores show comparable effects in both sexes does not rule out sex-specific effects.

(26) p.15: overrepresentation of close relatives: “Here we show that shared genetics most likely also play a substantial role”: I find this text to be overstated. Please either provide a quantification of the contribution of shared genetics to overrepresentation of close relatives, or change “substantial” to “greater than zero”.

We do not think that the original statement is an ‘overstatement’, but we do agree that being able to quantify the contribution would be better, which we have now done. For UKBB, we estimate that the direct effect of the genetic component to primary participation can account for 32% of the correlation of the liability scores between sibs that leads to a recurrent participation probability ratio of 2 (see p.23 in the main text).

(27) General comment: I suggest to define and use a single term for the trait being studied. The paper currently uses “participation”, “ascertained”, “ascertainment probability” interchangeably, but it would be better to use only one of these terms. I suggest “participation”

or “study participation”. I do not favor “ascertained” or “ascertainment probability”, as ascertainment is a term that has many different meanings and uses.

We understand the point of this comment, as we have struggled a little with this. The tricky thing is participation is not a trait like height. As we emphasized in the manuscript, it has a passive component, *i.e.* being invited which is commonly thought of as ascertainment bias, and an active component, *i.e.* agreeing to participate when invited, which is probably what most thought of as the participation trait. Our analysis of participation in UKBB would capture both of these effects. But, more generally, there would be sampling design where the bias is dominated by the passive component, while others would be dominated by the active component. When we use the different words, the intention is to put emphasis on different aspects of participation.

(28) General comment: Perhaps give the method a name? “DPO/DSIB1/DSIB20 method” (see above) may not be the best name.

Thanks for the suggestion. The ‘umbrella’ name/principle is shared versus not-shared comparison. Individually, DPO is now transmitted-nontransmitted-comparison, DSIB1 is within-sib-pair-comparison, and DSIB20 is between-sib-pair-comparison. Also noted is that the first two are comparing alleles in the same person, while the last is comparing genotypes of different individuals.

(29) General comment: this is not the first disease/trait analysis method to make use of shared genetic segments in related samples. Would it be of value to cite literature on linkage mapping (e.g. Ott et al. 2015 Nat Rev Genet) and population-based IBD mapping (e.g. Browning et al. 2012 Genetics)?

We agree that the idea here is related to what underlies linkage analysis. Instead of a pure IBD analysis like linkage analysis, ours is an IBD-based association/IBS analysis. We have added a sentence in the Discussion and referenced the affected relative pairs paper of Risch, given that our sib-pair analysis is essentially based on data of ‘affected’ sib-pairs.

Reviewer 2

This is a very important paper which addresses a very contemporary point in the discussion around GWAS studies. In particular the authors have developed a new methods for deriving a GWAS of participation without having measured non-participants but exploiting the IBD regions from sib pairs. The approach is very clever and of clear interest and usefulness.

We thank the reviewer for the kind words.

Although overall the work is fine and well conducted the paper is a bit difficult to follow especially in the methods. Clearly the authors have everything very clear in their heads but most readers will not. I think the paper would benefit for a bit more clarity in the explanation.

We agree with the reviewer, having focused most of our attention on introducing the ideas, some details on the applications of the ideas to real data were scant. This revision should remedy that. In particular, with the extra research, we now have a much better understanding of the nature of the major allele bias, which leads to substantial improvement of the analyses. In particular, we now have useful results on genetic correlations and heritability. Some details are given in the General Responses provided to all reviewers at the start of this document.

This said I do have a few comments for the authors:

(1) The authors should mention the explanation of why the IBD segments will be enriched in participation alleles compared to the non-shared ones. I went through the whole paper and found no mention of the simulations which are in the supplement apart from the legend of figure 9. The

authors should point this out in the main text otherwise their theorem seems postulated.

We agree. We now have Figure 4 in the main text that highlights the relationship between the frequency difference between the shared and not shared and the frequency difference between the sample and population. That quantitative relationship happens to be also needed to compute ‘effective sample size’ and estimate of heritability. The simulation conditions are simplified and clearly described in Methods. Similarities between effects captured by WSPC and BSPC are also investigated/discussed, including differences under a dominant or recessive model.

(2) Following up on the previous comment, the demonstration of the enrichment of participation alleles in IBD segments is extremely hard to follow as often new parameters are introduced without stating what they are. The reader has to deduce it from the rest of the text but it makes reading it very hard.

We appreciate that. As above, we believe that having Figure 4 in the main text, and the revised Methods section (simulations) on this would be a big improvement. Because it is a ‘new’ idea, our focus is first on testing, *i.e.* showing that the tests are insensitive to false positives resulting from sampling related confounding. We then use the UKBB data to show empirically that the method does indeed have power to detect participation signals. In the initial submission, the part on moving from testing to estimation was not given sufficient attention. We believe that this part of the manuscript is now substantially improved, in particular with the addition of the genetic correlations and heritability results.

(3) What does DPO stand for? I don't seem to be able to find any extended definition.

We thought of that as difference (D) between shared and not-shared alleles for parent-offspring (PO) pairs. In any case, we have now introduced new names to the three tests/comparisons which we hope are more descriptive. As before, they all fall under the umbrella of shared versus not-shared comparisons. Individually, we refer to them as transmitted-nontransmitted comparison (TNTC), within sib-pairs comparison (WSPC, the previous DSIB1), and between sib-pairs comparison (BSPC, the previous DSIB20). We further note that TNTC and WSPC are comparing alleles within the same individuals, while BSPC is comparing genotypes of different individuals/sib-pairs. This difference does not impact much on their ability to capture true participation effects, but is the main reason why TNTC and WSPC are much more sensitive to artefacts resulting from data and data processing errors.

(4) There is a similar problem with DSIB1 which is never defined anywhere.

Addressed in the response above.

(5) It would help if formulas were numbered.

Thanks for the suggestion. We now have the formulas numbered.

(6) I think something that is missing from the paper is an actual comparison between the method and running a GWAS of participation. I realise that just selecting out random samples from the UK biobank may not accomplish the task given that the authors postulate that the differences are embedded in the participants allelic frequencies, but at least simulations could be performed.

In Figure 4, through simulations, in addition to investigating what is described in our response to comment 1, we also examined the relation between the frequency difference of relative pairs and

‘unrelateds’ and the frequency difference between sample and population (and thus also the relationship between the former and the frequency difference between shared and not-shared within the relative pairs). These estimates are consistent with what are empirically observed with the UKBB data. In summary, we have a very good understanding of how participation bias can contribute to frequency difference in sampled relative pairs and ‘unrelateds’. Having said that, there are of course other factors that can contribute to frequency difference of relateds and unrelateds. While we believe that the conclusions that we have made in the manuscript based on our analyses are unlikely to be seriously impacted by these complications, examination of how participation bias can affect investigations of other phenomena is high on our list of future research.

(7) The study makes another strong assumption which is that the participating sib pairs or parent-offspring pairs are representative of the rest of UK biobank. Is this true? Although the authors say their results are unbiased from population stratification, this doesn't say anything about generalisability. It could be that these participants are more biased than the rest of UK biobank. This could be tested directly through GWAS of people with relatives vs the rest.

The response to the comment before also applies to this comment. In addition, we add the following. Firstly, the polygenic scores associations in Table 1, and the difference in PGS (with weights derived from one-half of the sib-pairs) values, between the other half of the sib-pairs and the unrelateds (p.21 in the revised manuscript), together show that our shared-versus-notshared analyses are capturing effects that apply to both the relative pairs and unrelateds. Having said that, we agree with the reviewer that performing a GWAS of people with relatives vs the rest is an obvious thing to do and indeed we have done so. The reason we currently did/do not highlight those results is because of a subtle but important statistical phenomenon. It happens that the shared versus not-shared comparison is not, in terms of sampling variation, uncorrelated with the

relative pairs versus unrelateds comparison. This is because a shared allele contributes twice to the overall allele frequency in the relative pairs, while a not-shared allele only contributes once. Without taking that into account one would find abnormally strong positive correlations between results from our GWAS and the relatives vs the rest GWAS. (That is why in the polygenic score comparison mentioned above we have to split the sib-pairs into two halves). In summary, the relatives vs the rest GWAS indeed should/does capture information that is useful for both adding to and comparing with the shared vs notshared GWAS. However, doing that properly requires extra research, especially since number of close relatives within the dataset is related to how many close relatives an individual has to begin with (i.e. an only child will not have a participating sibling), which is part of our ongoing effort.

(8) Looking at the frequency correction, it is unclear to me why the authors include only odd powers in the formula and why up to the power of 5.

First the odd power. In investigating the major allele bias, mathematically, we want our analysis/adjustment to be invariant to which allele of a SNP is coded as 1. In particular, we want our adjustments to be the same if for some SNPs, alleles 0 and 1 are switched. To achieve that, we regress the t-statistics on odd powers of centred allele frequency = (frequency of allele 1 – 0.5), and through the origin. Note that fitting through the origin is the same as not including the constant term, which can be considered as even power (power 0), in the regression. The regression on the odd powers of centred allele frequency assures that the fit/adjustment to frequency = a is negative of the fit/adjustment to frequency = 1-a. This also means there is no adjustment when allele frequency is 0.5. This is now clearly described in Methods. As to up to the fifth power, in the original submission, that was what we found to be significant in the fit, *i.e.* that provided a better fit than just having the linear term. In this revision, mainly because using *snipar* has greatly improved IBD estimation, we no longer need any adjustment to the BSPC

results, and adjustments using the linear term for centred allele frequency and the odd powers of minor allele frequency are sufficient for TNTC and WSPC.

(9) Why were quantitative traits standardised separately? This may induce bias in the results as they will refer to $1SD$ which corresponds to a different number in the two sexes. For example for BMI women have a higher variance than men and thus dividing the values by these numbers will be effectively moving the data from the same scale (BMI) to two distinct ones which may influence the results.

While, depending on the circumstances, there could be reasons to standardize the sexes jointly as opposed to separately, our default preference is for the latter. Most importantly, we do not think that would lead to false positives in our analyses. However, we do take the comment of the reviewer seriously, and have performed analyses with the two sexes standardized jointly, and observed no meaningful difference in the results/conclusions.

(10) The authors have used only 40 PCs to correct for population stratification. Previous studies have shown that this is not enough (see for example PMID: 31636407). They should either use more or use Grammar-Gamma residuals as implemented in GCTA for example.

First, because of the ‘case-control’ matching in our GWAS, population stratification should not contribute to the signals detected. (The major allele bias is a consequence of genotyping and phasing errors and not related to PCs). In the analyses of the ‘unrelateds’, we believe that adjusting for 40PCs are sufficient to demonstrate our main point that the shared versus not-shared analyses can and do capture information on the direct genetic effect to participation bias, i.e. the observed association between the polygenic scores and the traits in the unrelateds is not driven by population stratification confounding. Some earlier work of ours and others have

shown that the higher order PCs are often just capturing noise (Young *et al*, Deconstructing the sources of genotype-phenotype associations in humans *Science* **365**, 1396-1400, 2019). Also, we are beginning to believe that, analyses of ‘unrelateds’ only, *i.e.* without family controls, can never guarantee the elimination of population stratification and/or indirect genetic effects (Young *et al*, Mendelian imputation of parental genotypes improves estimates of direct genetic effects. *Nature Genetics* **54**, 897-905, 2022).

Reviewer 3

(1) The subject of this paper is interesting and important – that participation in genetic or other studies is a heritable trait with many phenotypic and genetic associations. The abstract slightly overstates some of the background- that it is not possible to examine participation directly (it is, studies such as UK Biobank have looked at later optional components) and these studies (as well as studies looking at the phenotypic associations of participation) have made similar discoveries

We thank the reviewer for the comments. Before addressing this first point, we want to recommend the reviewer to read the General Responses at the start of this document. We agree that the UKBB data have substantial ascertainment bias was well established based on known differences in variables such as educational attainment. We also acknowledge that genetics of participation have been studied through GWAS performed based on what we referred to as secondary participation events. Our contributions are, however, quite different. Our manuscript is submitted as a Technical Report. We start by showing that there is information in the sampled genotypes themselves, without phenotypes, that captures the bias of participation on a genetic level. This information, as far as we know, has never been utilized in this manner for this purpose (although the principle of why it works is related to other methods such as the transmission-disequilibrium-test and affected sib-pair linkage analysis). To us, this is the main ‘discovery’. Secondly, not only have we ‘discovered’ extra information, our information is different in nature from that utilized by existing approaches. For example, GWAS performed for the secondary participation events are based on a sample that is biased to start with, and the genetic component to secondary participation, while expected to be substantially correlated with that for primary participation, is probably not the same. The new results we have (Table 2 and Extended Data Table 3) on genetics correlation and heritability reinforce this point. Having said that, we emphasize that we do not intend to say other approaches to studying participation are not

important. Our point is the new information we discovered can add on to results obtained from other analyses. We now explicitly make this last point in Discussion (end of second paragraph in Discussion).

(2) The abstract contains several results with p-values in parenthesis after the text, but no effect size. This is not best practice – there should be effect sizes (R², or standardised betas) accompanying each p-value, to aid interpretation. The polygenic score is for participation (participation implicitly coded 1, non-participation = 0) – but that is not clear in the abstract.

Giving the p-values is to emphasize that the ‘novel’ information we discovered is genuinely capturing useful information about participation. It is not really about using the polygenic score as a predictor of these other variables. We do agree that effect size is important, but effect size of polygenic scores that only capture a small fraction of the overall genetic component is not necessarily the best to highlight in the abstract. Our new results on genetic correlations (Table 2) and heritability should address the effect size problem. We have added the estimated genetic correlation between primary participation and educational attainment (36.7%) to the Abstract.

(3) The examples given (presidential candidate voting, vaccination status) are vague and it isn't made clear in the next sentence what hypothesis is being tested in these 'fictional studies' and what is hypothetically being adjusted for in the next.

The point of those examples are not about specific details, but if those details are of interest they can be found in references 1-3 in our paper. We make these points in the manuscript because the serious mis-predictions of these events, i.e. the overestimation of US vaccine uptake (see reference 1) and the misleading presidential polls (see reference 2 and 3), are something that an average person can relate to. The fact that much resources and expertise were involved in the US

presidential election polling is also something we believe people can appreciate without knowing the technical details. This highlights the difficulty of the ascertainment bias problem, arguably the most difficult in applied statistics. We are not sure what the reviewer is referring to as ‘fictional studies’, given that reference 1 is a recent article in Nature on the overestimation of US vaccine uptake, and reference 2 is a BBC article lamenting the failures of presidential polling. The latter include the 2016 US presidential election when many who followed the polling results closely were surprised by the result.

(4) I understand the genetic principles set out in the introduction, that alleles associated with participation will be more common in segments that are present in multiple participants.

a) I do not follow why this would necessarily be immune to population stratification.

b) Furthermore, alleles correlated between segments on different chromosomes of the same individual are known to be correlated when there is assortative mating in a population. This has been demonstrated for educational attainment, with which they find a strong association. How can we be confident that the results shown do not represent the effects of assortative mating within a population?

a. The reason that population stratification would not lead to false positive is similar to that behind TDT, the transmission-disequilibrium- testing. Specifically, in the case of TNTC and WSPC, we are actually comparing alleles within the same person, so any population stratification effect cancels. In the case of BSPC, it is not as obvious but still true, because as noted, assuming no selection effect, a sib-pair has an equal chance (1/4) to be IBD0 or IBD2 at a particular locus. b. Yes, two variants on two different chromosomes correlated because of assortative mating is a good case to examine. Suppose allele 1 at locus A is positively correlated with allele 1 at locus B on another chromosome in the parents. There is the correlation of the alleles within a parent, and there is correlation between the alleles in the father and those in the

mother. For simplicity, assume the correlation is the same within or between parents, which would be the case if this correlation is a result of assortative mating in equilibrium, i.e. over many generations. Suppose allele 1 in locus A has a direct positive effect in participation, but not locus B. Allele 1 of locus A will be enriched in the parents of the participating sib-pairs (relative to population) and so would allele 1 of locus B although to a lesser degree. This will also be the case for alleles in the participating sib-pairs themselves. Indeed, that is the typical confounding effect assortative mating can induce in the usual control-control analysis. However, because they are on different chromosomes, transmissions at the two loci, and so are IBD sharing, are independent of each other. Suppose a parent is heterozygote at both locus A and locus B. Conditioning on the two sibs both participating, we will see allele 1 at locus A transmitted more often to the siblings/offsprings, and more likely to be a shared allele than allele 0. But that does not affect the transmission and consequentially the shared versus not shared difference in locus B, given that the transmissions are independent at the two loci. In summary, relative to the population, allele 1 at locus B will be enriched in the participating sib-pairs, but there is no enrichment of allele 1 in the shared alleles relative to the not-shared alleles.

(5) I assume from following the definition of the statistics that the F terms are the allele frequencies in the shared/non-shared segments (FPT = frequency of the 1-coded allele in segments transmitted from parents to offspring; $FPNT$ = frequency of the 1-coded allele in segments not transmitted from parents and offspring). Could this be pointed out more explicitly in the text as an aid to readers?

Yes, and we are now calling the previous DPO test the transmitted-nontransmitted-comparison (TNTC), so that should be more clear.

(6) Is the derivation of the standard error for the DPO, DSIB1, and DSIB20 statistics (referred to on Line 206) given anywhere?

Thanks for pointing out this omission in the original submission, which happens to be relevant to some of the issues discussed above. We started off calculating variances assuming independence of the parental alleles, *i.e.* HWE. We then realized that this is both unnecessary and suboptimal. What we do now, in the revised manuscript, is simpler and more accurate. It is described clearly in the Methods now. In particular, for TNTC and WSPC, we performed a one-sample t-test in the elementary statistics sense. Take WSPC as the example, for each pair i , we calculated

$$D_i = \text{shared allele} - (\text{sum of the two notshared alleles})/2.$$

Assuming there are n pairs, the SE of the sample average of \bar{D} is

$$\frac{SD(D_i)}{\sqrt{n}}$$

where $SD(D_i)$ is the sample standard deviation of the D_i s. Essentially the only

assumption here is the D_i s are independent. For the BSPC, it is a two sample t-test without assuming any relationship between the variances of the two frequency averages being compared.

(7) The major allele bias seems largely unexplained and, without a reasonable explanation. This raises the general possibility that, in the absence of a strong/proven explanation for this and the multiple ‘spurious’ MHC variants – other sources of bias may be lurking? For example, would a small proportion of minor allele homozygotes being miscalled as heterozygotes lead to such bias? The authors also state that “part of this ‘major allele effect’ might be real” (Lines 184-185). Can they explain what real processes would lead to this effect? This paper would perhaps be better suited to a more technical journal where these issues can be addressed.

Firstly, the reviewer is referred to the General Responses provided at the start of this document. We have spent most of the time in the last 10 months understanding the sources of this bias, and

deriving solutions. Most importantly, while TNTC and WSPC, in the step that splits a genotype into the shared and notshared allele, are affected by genotyping and phasing errors that lead to an artificial major allele bias, the BSPC results look very clean, e.g. having a fitted intercept of practically 1 when LD score regression is applied. However, the BSPC results still have a small positive correlation with allele frequency, although that is negligible compared to that of WSPC ($r^2 = 5.6 \times 10^{-5}$ compared to 0.030). This small correlation, or part of it, can indeed be ‘real’. Firstly, there is no reason to believe that the effects of the major and minor alleles would be perfectly balanced among the set of SNPs studied. Secondly, and most interestingly, if we assume the effects of the major and minor alleles are perfectly balanced in the population, alleles that have a higher frequency in the sample would actually have a slightly higher chance of having a positive participation effect than alleles that have a lower frequency. This is because selection would increase the frequency of an allele with positive participation effect relative to the population. This is most easily seen with a SNP that has 50% frequency for both alleles in the population. The positively selected allele would have a frequency higher than 50% in the sample, and appears to be the major allele. That means selection bias would actually lead to a major allele bias, although that effect is expected to be very small.

(8) Why is it necessary to GC-adjust? This is a now infrequently used approach in GWAS, leading to over-correction when the signal is polygenic. What is the justification for using it in the current circumstances?

Firstly, we no longer make any adjustment to the BSPC results. We do apply now (different from before) a minor-allele-frequency (MAF) specific genomic control to the TNTC and WSPC results (see Methods). That is because we find that, due to genotyping and phasing errors, errors in splitting the genotype into the shared and notshared alleles lead to inflation of the chi-square statistics. Importantly, the average inflation is much higher for SNPs with low MAF probably

because their genotypes tend to have a higher error rate. This means, without adjustment, the chi-square statistics are negatively correlated with MAF. As MAF is substantially positively correlated with LD scores, applying LD score regression to the unadjusted chi-square statistics would lead to nonsensical results. The MAF-specific genomic control adjustment removes the correlation between the chi-square statistics and MAF. See the General Responses and Methods for more details.

(9) It's not clear how the authors chose to present some traits that were not associated with pPGS and not others. Why not apply a PheWAS approach and provide a plot that allows the reader to look at phenotype domains and how each might be impacted by participation bias? Similarly, UKBB has other optional assessments (aide-memoire form, Imaging visit, Mental Health Questionnaire, Healthy Work questionnaire, Cognitive function online follow-up) in addition to the diet-by-recall and the physical activity monitoring that were reported in the study. Were these two follow-up assessments analysed for a specific reason or could the analysis be expanded to include the other assessments? Being more comprehensive would make the present study easier to compare to previous studies on the genetics of optional participation in UKBB.

To us, we are presenting a new method. In addition to theory and simulations, we want to show that it works with real data. For the latter, figuring out the major allele bias caused by data errors took up much of our attention in the past 10 months. We believe the manuscript has enough novel observations in the analyses, particularly now with the genetic correlation results and estimate of heritability. Being comprehensive with the traits is not a priority to us here. Each trait and their data could have their own idiosyncrasy and we believe that those who know and care about specific traits should apply the method themselves. Indeed, it is our hope that examining primary participation associations will be a standard procedure when performing GWASs in the future.

(10) *Minor issues: A) This sounds awkward: ‘identifying such (A) component’ [abstract].*

B) Code available on publication – but not now, so that reviewers can spot errors?

A) We change to “Identifying this component”. B) Like many others, we cannot guarantee that our code is perfect. However, we are responsible if the results we presented are misleading due to errors in the code. But the same is true with other errors. Computer code tends to improve overtime. Also, the implementation of our ideas has improved substantially over time, e.g. the use of *snipar* in the place of KING. Our preference is to release software when the manuscript is formally published.

Decision Letter, first revision:

13th Feb 2023

Dear Stefania,

Your Technical Report, "The Genetics of Participation: Method and Analysis" has now been seen by the original 3 referees. You will see from their comments below that while they continue to find your work of interest and find it has improved in revision, there are still some outstanding points that require resolution. We are interested in the possibility of publishing your study in Nature Genetics, but would like to consider your response to these concerns in the form of a revised manuscript before we make a final decision on publication.

Briefly, Referees #2 and #3 are now satisfied and supportive of publication. Conversely, Referee #1 - while appreciating the improvement in the simulations - does not believe that these added results have satisfactorily addressed their original comment #1 from the first round of review. As this criticism is directed at the fundamental reliability and interpretation of your proposed method, we believe it must be fully clarified before a final decision can be made.

To guide the scope of the revisions, the editors discuss the referee reports in detail within the team, including with the chief editor, with a view to identifying key priorities that should be addressed in revision and sometimes overruling referee requests that are deemed beyond the scope of the current study. We hope that you will find the prioritized set of

referee points to be useful when revising your study. Please do not hesitate to get in touch if you would like to discuss these issues further.

We therefore invite you to revise your manuscript taking into account all reviewer and editor comments. Please highlight all changes in the manuscript text file. At this stage we will need you to upload a copy of the manuscript in MS Word .docx or similar editable format.

*2) If you have not done so already please begin to revise your manuscript so that it conforms to our Technical Report format instructions, available [here](http://www.nature.com/ng/authors/article_types/index.html). Refer also to any guidelines provided in this letter.

[redacted]

We hope to receive your revised manuscript within four to eight weeks. If you cannot send it within this time, please let us know.

Nature Genetics is committed to improving transparency in authorship. As part of our efforts in this direction, we are now requesting that all authors identified as 'corresponding author' on published papers create and link their Open Researcher and Contributor Identifier

(ORCID) with their account on the Manuscript Tracking System (MTS), prior to acceptance. ORCID helps the scientific community achieve unambiguous attribution of all scholarly contributions. You can create and link your ORCID from the home page of the MTS by clicking on 'Modify my Springer Nature account'. For more information please visit www.springernature.com/orcid.

Sincerely,

Michael Fletcher, PhD
Senior Editor, Nature Genetics

ORCID: 0000-0003-1589-7087

Reviewers' Comments:

Reviewer #1:

Remarks to the Author:

The concerns have generally been addressed, and the manuscript is much improved. A few comments remain.

Major comment:

1. Reviewer #1 comment 1 requested that "the simulations should check that there is a 100% genetic correlation between the DPO/DSIB1/DSIB2 [now called TNTC/WSPC/BSPC] association statistics from cases and case-control association statistics". Reviewer #2 comment 6 requested "an actual comparison between the method and running a GWAS of participation". However, the simulations provided (p.19-20 and Figure 4) only partially address these requests. Specifically, I am not sure how to interpret that the finding that "the two ratios are approximately 0.86, meaning that frequency differences between the shared and not-shared [sic] in sibling pairs are about 86% of the frequency differences between the alleles in participants and the population" (p.20).

If the factor of 0.86 implies that all of the results (estimated effect sizes, genetic correlations, heritability estimates etc.) are expected to be biased by a factor of 0.86 (or other values at different λ_S ; Figure 4), then the manuscript must note this bias as an explicit caveat.

If despite the factor of 0.86 the authors believe that all of the results (estimated effect sizes, genetic correlations, heritability estimates etc.) are unbiased, then the manuscript must provide evidence that these results are unbiased in simulations. This evidence has not been provided.

Also: the frequency difference between participants and the population (denominator used in the manuscript) differs slightly from the frequency difference between participants

(cases) and controls, which is the quantity analyzed in a GWAS. The manuscript should comment on this distinction.

Minor comments:

p.1: I do find "Studying the genetics of participation using samples only" to be preferable to the current title. Even better would be "Genetic architecture of study participation using data from participants only" (as "samples" is ambiguous).

p.1: "with a gene-environment interaction component": this paper does not seem to include any analysis that demonstrates a gene-environment interaction. Either an analysis that demonstrates a gene-environment interaction should be included, or the claim about the presence of a gene-environment interaction should be deleted from the Abstract.

p.2-7: I find it odd to not include an explicit heading distinguishing the Introduction section (providing broad background) from the Methods overview section (introducing the method). It would be trivially easy to add such an explicit heading.

p.7: the text describing the TNTC, WSPC, BSPC tests is appropriate. However, please add to p.7 (and/or p.10) both (a) text describing how the actual test statistics are computed (can be copied from line 616-617 on p.29 of Methods), and (b) text describing which test statistics are provided as input to LD score regression (e.g. distinguishing between t-statistics vs. chisq statistics). This content is critical to the main results of the paper, and must be described in Methods overview in main text.

p.7: "having the individual results separately is important because they could capture different effects depending on the nature of the ascertainment bias": I suggest that the manuscript should state explicitly whether TNTC, WSPC, BSPC capture different effects w.r.t. "voluntary" vs. "involuntary" effects mentioned in the previous sentence. If yes, please explain why. If no, then it is odd to mention "voluntary" vs. "involuntary" effects and then state that TNTC, WSPC, BSPC could capture different effects.

p.19-20 and Figure 4: I was expecting to see Figure 4 describe simulation results related to $F_T - F_{NT}$ (TNTC test, Equation 1) and $F_{IBD1S} - F_{IBD1NS}$ (WSPC test, Equation 2) and $F_{IBD2} - F_{IBD0}$ (BSPC test, Equation 3). However, I instead see simulation results related to $(F_{SIBS} - F_{SING})$ (???) and $F_{IBD1S} - F_{IBD1NS}$ (WSPC test, Equation 2) and $F_{IBD2} - F_{IBD0}$ (BSPC test, Equation 3). Why $F_{SIBS} - F_{SING}$ (???) instead of $F_T - F_{NT}$ (TNTC test, Equation 1)? It is odd for the simulations figure to deviate from Equations 1,2,3 in this way, and this requires an explanation/justification (or, replace $F_{SIBS} - F_{SING}$ with $F_T - F_{NT}$).

p.22: given that Table 1 includes PGS correlation results for TNTC and WSPC and BSPC and Combined, and that genetic correlation results are very closely related to PGS correlation results, please include and cite genetic correlation results for TNTC and WSPC and BSPC and Combined within a single table (either Table 2 or Extended Data Table 2).

General comment: I find the ordering of the results to be odd. In particular, the paper presents PGS correlation results in real data on p.11-17 (Table 1 and Figure 4), then presents simulation results on p.18-21 (Figure 4), then presents genetic correlation results

(which are very closely related to PGS correlation results, except more principled and more interpretable; see Reviewer 1 Comment 4 and Reviewer 3 Comment 2) in real data on p.21-22 (Table 2). It is odd to sandwich two very closely related real data analyses with simulations in between. Typically the simulations would be presented either before all of the real data results (most common), or after all of the real data results.

Reviewer #2:

Remarks to the Author:

The paper has greatly improved and is now more precise.

I now have a much better understanding of what they have done.

In fact, I would have had additional comments which however have been addressed thanks to the other reviewers.

I think that further work needs to be conducted in the future to improve this method, especially in the IBD inference part.

However, I feel that the paper is an interesting contribution as it is and it would merit publication.

Reviewer #3:

Remarks to the Author:

I think the authors have addressed all of the reviewers' comments very thoroughly. Apart from one figure (the flow diagram, with multiple 'blobs' and colours), which might be better placed in the supplementary information, I think the manuscript is suitable for publication.

Author Rebuttal, first revision:

Reviewer 1:

The concerns have generally been addressed, and the manuscript is much improved. A few comments remain.

Major comment:

(1) Reviewer #1 comment 1 requested that "the simulations should check that there is a 100% genetic correlation between the DPO/DSIB1/DSIB2 [now called TNTC/WSPC/BSPC] association statistics from cases and case-control association statistics". Reviewer #2 comment 6 requested "an actual comparison between the method and running a GWAS of participation". However, the simulations provided (p.19-20 and Figure 4) only partially address these requests.

Specifically, I am not sure how to interpret that the finding that "the two ratios are approximately 0.86, meaning that frequency differences between the shared and not-shared [sic] in sibling pairs are about 86% of the frequency differences between the alleles in participants and the population" (p.20).

If the factor of 0.86 implies that all of the results (estimated effect sizes, genetic correlations, heritability estimates etc.) are expected to be biased by a factor of 0.86 (or other values at different λ_S ; Figure 4), then the manuscript must note this bias as an explicit caveat.

If despite the factor of 0.86 the authors believe that all of the results (estimated effect sizes, genetic correlations, heritability estimates etc.) are unbiased, then the manuscript must provide evidence that these results are unbiased in simulations. This evidence has not been provided.

Also: the frequency difference between participants and the population (denominator used in the manuscript) differs slightly from the frequency difference between participants (cases) and controls, which is the quantity analyzed in a GWAS. The manuscript should comment on this distinction.

We thank the reviewer for the comments.

In the first revision, we believed that we had performed the work required to address the comments of Reviewer 1 and Reviewer 2 noted above. Reading the comments from Reviewer 1 here, it is evident that we were not clear, explicit, or direct enough in our presentation and responses. We have therefore made a second revision of the manuscript that we hope would clarify these issues. Specifically,

- a. We do not think of the 0.86 factor as a ‘bias’. Instead, the factor shows how two closely related, but not identical, parameters are related. Another related but a more obvious example is described in point c below. Importantly, 0.86 is a scaling factor, *i.e.* it is multiplicative and not additive. In particular, this means that the change in allele frequency from population to sample has the same sign as the difference in frequency between the alleles only in one individual of a participating relative pair and the alleles that are in both members of a participating relative pair. This is precisely what we refer to as the First Principle of Genetic Induced Ascertainment Bias. This property alone is sufficient for testing purposes and for constructing polygenic scores that have predictive power.
- b. The multiplicative factor 0.86 applies (approximately) to all variants that have small individual effects under an additive model. Importantly, as the simulations show, the same factor applies to both WSPC and BSPC (Figure 4). (TNTC is addressed below in point d). This means that the genetic correlation is close to 100% between WSPC and BSPC. Also, because it is the same multiplicative factor, regardless of what it is, it does not affect the relative weights of the polygenic risk scores (results in Table 1 and Figure 3), and there is also no impact on the genetic correlation estimates (results in Table 2).
- c. The factor 0.86 does affect the heritability estimate. In the last part of the Methods section (previous and current revisions), we described how this factor was used to calculate the heritability estimate based on the BSPC results. The same part of Methods gave the simple relationship between frequency difference of sample versus population and that of sample versus nonparticipants:

$$(F_{samp} - f_{pop}) = (1 - \alpha)(F_{samp} - F_{nonparticipants}), \text{ where } \alpha \text{ is the participation rate.}$$

Notably, here is another multiplicative factor of $(1 - \alpha)$ linking two frequency differences. This reinforces the point that there is nothing special about having a multiplicative factor linking $(F_{IBD2} - F_{IBD0})$ and $(F_{samp} - f_{pop})$.

- d. Here, we address the point about the potential difference between TNTC and WSPC/BSPC. The WSPC and BSPC empirical results and simulations are based on the same set of sibling pairs, *e.g.* a sibling pair would in general be IBD0, IBD1, and IBD2 at different sites. The results and selection are thus driven by the same effects of variants on individuals. Hence it is meaningful to use the simulations (or mathematical calculations) to study the relationship between the WSPC and BSPC results. TNTC, however, is based on a different set of samples. If we perform simulations assuming the same model, including the same effects for variants, then we can expect that the genetic component captured/estimated to be essentially the same as that of WSPC. However, given that, among other differences, the parents in the parent-offspring pairs are older and the relationship between a parent and an offspring is asymmetric, that assumption is questionable. For UKBB, the average year of birth of the sibling pairs (1950.8) is very close to that of the WB sample as a whole (1951.2). By contrast, the parents of the parent-offspring pairs have a much earlier average (1941.7), and the offspring have a much later average (1965.0). Adding this to having larger sample size and being insensitive to data artefacts, genetic correlation and heritability estimates based on BSPC are much more relevant than that based on TNTC. However, regardless of how correlated the genetic component captured by TNTC is correlated with that of WSPC and BSPC, which in general could depend on many factors including the recruitment scheme, the TNTC results for the UKBB data were shown to have predictive power through the polygenic score analyses (Table 1). That supports the First Principle that we introduce in the manuscript. Related to some of the minor comments below, the discussion here also highlight a difference between polygenic score prediction and estimation of genetic correlation and heritability.
- e. While Figure 4 shows that WSPC and BSPC are essentially capturing the same effects under the additive model, we also had performed simulations for dominant and recessive

models. The simulations show (lines 396-398 in the first revision, lines 403-406 in this revision) that the BSPC effects would be about 10% smaller than that of WSPC for a variant with a dominant effect for the positive-effect allele, and about 8% higher with a recessive model. Hence, in practice, deviations from the additive model could reduce the genetic correlation between BSPC and WSPC somewhat, but the impact is likely to be small.

In this revision, the main text had been modified (lines 406-412, lines 472-480) to clarify the issues discussed here. Line 478 gives the relationship

$(F_{samp} - f_{pop}) = (1 - \alpha)(F_{samp} - F_{nonparticipants})$ noted in point c. Furthermore, Supplementary

Note Section 6 is added to elaborate on point d.

Minor comments:

(2) p.1: I do find "Studying the genetics of participation using samples only" to be preferable to the current title. Even better would be "Genetic architecture of study participation using data from participants only" (as "samples" is ambiguous).

We thank the reviewer for this suggestion. We would also suggest to the editor: "Studying the genetics of participation using footprints left on the ascertained genotypes", or "Studying the genetics of participation using data from participants only".

(3) p.1: "with a gene-environment interaction component": this paper does not seem to include any analysis that demonstrates a gene-environment interaction. Either an analysis that demonstrates a gene-environment interaction should be included, or the claim about the presence of a gene-environment interaction should be deleted from the Abstract.

In the abstract, we have now changed ‘with a gene-environment interaction component’ to ‘with a genetic component’.

(4) p.2-7: I find it odd to not include an explicit heading distinguishing the Introduction section (providing broad background) from the Methods overview section (introducing the method). It would be trivially easy to add such an explicit heading.

The heading has now been added.

(5) p.7: the text describing the TNTC, WSPC, BSPC tests is appropriate. However, please add to p.7 (and/or p.10) both (a) text describing how the actual test statistics are computed (can be copied from line 616-617 on p.29 of Methods), and (b) text describing which test statistics are provided as input to LD score regression (e.g. distinguishing between t-statistics vs. chisq statistics). This content is critical to the main results of the paper, and must be described in Methods overview in main text.

To comply with the reviewer’s suggestion, the following sentence has been added to the main text (lines 178-179):

For each sequence variant, t-statistics for TNTC, WSPC and BSPC were computed by dividing each of the frequency difference, expressions (1) to (3), by its standard error (SE).

Furthermore, we have changed ‘BSPC GWAS results’ to ‘ χ^2 statistics computed from BSPC’ in the following sentence (lines 200-204):

For example, when LD score regression (LDSC²⁰) is applied to the χ^2 statistics computed from BSPC, the fitted intercept is nearly exactly 1 (0.9998), indicating that the χ^2 statistics are neither inflated because of data artefacts, nor are they affected by issues such as population stratification.

(6) p.7: "having the individual results separately is important because they could capture different effects depending on the nature of the ascertainment bias": I suggest that the manuscript should state explicitly whether TNTC, WSPC, BSPC capture different effects w.r.t. "voluntary" vs. "involuntary" effects mentioned in the previous sentence. If yes, please explain why. If no, then it is odd to mention "voluntary" vs. "involuntary" effects and then state that TNTC, WSPC, BSPC could capture different effects.

In terms of real effects, point d of our response to the major comment has addressed the issue: WSPC and BSPC are capturing very similar effects while TNTC could be something more different. The voluntary and involuntary effects are both real effects. E.g. we would not expect the relationship between TNTC and WSPC/BSPC to be the same in studies with very different recruitment schemes. Regardless of their relative contributions, in a given study, WSPC and BSPC would be very similar as they are based on the same set of sibling pairs, and each can contribute to the difference between TNTC and WSPC/BSPC. We note that having individual results separately is not just because real effects can be different. In the manuscript we have emphasized that having separate results is especially important because data artefacts affect the three comparisons in a different manner. In particular, BSPC is very robust while TNTC and WSPC are more sensitive to data problems as they require splitting a genotype into shared and not-shared.

In summary, multiple factors can contribute to differences between the three comparisons, both real effects and data artefacts. To make this clearer in the manuscript, we have revised the text that the reviewer mentions (lines 157-160) and it now reads as follows:

However, having the individual results separately is important because they could capture different effects depending on the nature of the ascertainment bias, which is in part determined by the recruitment scheme. Most importantly, as demonstrated below, they are impacted differently by genotyping and data processing errors.

In addition, as noted in our response to the major comment, we have added a Supplementary Note Section 6 to elaborate on this (noted on line 412 of the main text).

(7) p.19-20 and Figure 4: I was expecting to see Figure 4 describe simulation results related to $F_T - F_{NT}$ (TNTC test, Equation 1) and $F_{IBD1S} - F_{IBD1NS}$ (WSPC test, Equation 2) and $F_{IBD2} - F_{IBD0}$ (BSPC test, Equation 3). However, I instead see simulation results related to $(F_{SIBS} - F_{SING})$ (???) and $F_{IBD1S} - F_{IBD1NS}$ (WSPC test, Equation 2) and $F_{IBD2} - F_{IBD0}$ (BSPC test, Equation 3). Why $F_{SIBS} - F_{SING}$ (???) instead of $F_T - F_{NT}$ (TNTC test, Equation 1)? It is odd for the simulations figure to deviate from Equations 1,2,3 in this way, and this requires an explanation/justification (or, replace $F_{SIBS} - F_{SING}$ with $F_T - F_{NT}$).

The reason for not including $(F_T - F_{NT})$ was addressed in point d of our response to the major comment. That is, if we were to simulate parent-offspring pairs with the same model assumptions and effect sizes, TNTC would be similar to WSPC as parent-offspring pairs share one allele IBD at every locus. We include $(F_{SIBS} - F_{SING})$ in the figure because of our Second Principle. While more sensitive to confounding, $(F_{SIBS} - F_{SING})$ captures additional

information on participation bias on top of TNTC/BSPC/WSPC and by including it in the figure, we show how the First and Second Principles relate to one another.

(8) p.22: given that Table 1 includes PGS correlation results for TNTC and WSPC and BSPC and Combined, and that genetic correlation results are very closely related to PGS correlation results, please include and cite genetic correlation results for TNTC and WSPC and BSPC and Combined within a single table (either Table 2 or Extended Data Table 2).

We agree that, in theory, PGS correlation and genetic correlation are closely related. However, in practice, PGS correlation with traits and the estimate of genetic correlation using LDSC are very different when it comes to sensitivity to the data artefacts we highlighted. In particular, while the polygenic scores predictions are very robust, i.e. insensitive to the artefacts, that is not the case when we apply LD score regression to the TNTC and WSPC statistics. In particular, we have noted that for TNTC and WSPC, the data artefacts create a major allele bias to the t-statistics, and a MAF-related bias for the chi-square statistics. Because MAF is correlated with LD-scores, this creates a problem with the heritability estimate of LDSC and through that, also affects genetic correlation estimates. In particular, without the MAF-adjustment that we applied to the chi-square statistics, both TNTC and WSPC would have a negative heritability estimate from LDSC, and hence no estimates of genetic correlation with other traits. With the major allele MAF-adjustment, WSPC gives some more reasonable results, but that does not mean that the effect of the artefacts are completely eliminated with respect to estimating genetic correlation using LDSC. There is some discussion of that in Methods. At this moment, because BSPC is not affected by the artefacts, we prefer to only highlight the BSPC estimates in the main results. We did/do provide the WSPC-based genetic correlations in Extended Data Table 2, but that is mainly to show that our adjustments have reduced the impact of the artefacts. Giving more genetic estimates using WSPC and/or TNTC at this point could give credit to results that do not deserve

them. To make this clear, we have now added the following sentence to the main text (line 452-453):

Further research, however, is needed to determine the best way to use WSPC and TNTC results for genetic correlation estimates.

(9) General comment: I find the ordering of the results to be odd. In particular, the paper presents PGS correlation results in real data on p.11-17 (Table 1 and Figure 4), then presents simulation results on p.18-21 (Figure 4), then presents genetic correlation results (which are very closely related to PGS correlation results, except more principled and more interpretable; see Reviewer 1 Comment 4 and Reviewer 3 Comment 2) in real data on p.21-22 (Table 2). It is odd to sandwich two very closely related real data analyses with simulations in between. Typically, the simulations would be presented either before all of the real data results (most common), or after all of the real data results.

This question is partially addressed by the answer to the previously question. Here, we add the following. We use the PGS results to validate the First Principle and its application. It is thus important to present them as early as possible. We prefer to introduce these UKBB results before the simulations, because the simulations were performed under model assumptions that were chosen with the specific circumstances of the UKBB in mind. While the PGS results are results of statistical testing, the genetic correlations and heritability estimates are results of statistical estimations. Although related, the latter require quite a bit more modelling and assumptions. Indeed, as noted in the response to the major comment, the simulation results ‘presented in between’ are needed for us to perform the estimations. In particular, the 0.86 factor is needed for the heritability estimate. And demonstrating that the 0.86 factor applies to SNPs of various effect sizes are needed for the genetic correlation estimate.

Reviewer 2:

The paper has greatly improved and is now more precise.

I now have a much better understanding of what they have done.

In fact, I would have had additional comments which however have been addressed thanks to the other reviewers.

I think that further work needs to be conducted in the future to improve this method, especially in the IBD inference part.

However, I feel that the paper is an interesting contribution as it is and it would merit publication.

We thank the reviewer for the kind words.

Reviewer 3:

I think the authors have addressed all of the reviewers' comments very thoroughly. Apart from one figure (the flow diagram, with multiple 'blobs' and colours), which might be better placed in the supplementary information, I think the manuscript is suitable for publication.

We thank the reviewer for the kind words and for the comment. We believe that the flow diagram is more suitable in the main text as this is paper is a technical report.

Decision Letter, second revision:

16th Mar 2023

Dear Stefania,

Thank you for submitting your revised manuscript "The Genetics of Participation: Method and Analysis" (NG-TR59513R1). It has now been seen by the original referees and their comments are below. The reviewers find that the paper has improved in revision, and therefore we'll be happy in principle to publish it in Nature Genetics, pending minor revisions to satisfy the referees' final requests and to comply with our editorial and formatting guidelines.

Sincerely,

Michael Fletcher, PhD
Senior Editor, Nature Genetics

ORCID: 0000-0003-1589-7087

Reviewer #1 (Remarks to the Author):

I am satisfied with the minor revisions in response to my comments, and I recommend publication.

I have two very minor comments that do not require re-review:

1. The authors might add a sentence somewhere commenting on the fact that the frequency difference between participants and the population (denominator used in the manuscript) differs slightly from the frequency difference between participants (cases) and controls, which is the quantity analyzed in a GWAS (see end of Reviewer #1 Major comment 1 in second round of review).
2. The authors might add a sentence at the beginning of the paragraph of line 369 providing an overview of the motivation for this set of analyses, as this would further improve clarity.

Author Rebuttal, second revision:

Point by point answers to reviewers

Benonisdottir and Kong

Studying the genetics of participation using footprints left on the ascertained genotypes

Reviewer 1:

I am satisfied with the minor revisions in response to my comments, and I recommend publication.

I have two very minor comments that do not require re-review:

1. The authors might add a sentence somewhere commenting on the fact that the frequency difference between participants and the population (denominator used in the manuscript) differs slightly from the frequency difference between participants (cases) and controls, which is the quantity analyzed in a GWAS (see end of Reviewer #1 Major comment 1 in second round of review).

To comply with the reviewer's comment we have added the following to the main text:

Results highlighted here are frequency differences relative to $(F_{samp} - f_{pop})$.

The latter has the following relationship with the frequency difference between the participants and non-participants,

$$\left(F_{samp} - F_{nonparticipants}\right),$$

$$(F_{samp} - f_{pop}) = (1 - \alpha)\left(F_{samp} - F_{nonparticipants}\right). \#(4)$$

2. *The authors might add a sentence at the beginning of the paragraph of line 369 providing an overview of the motivation for this set of analyses, as this would further improve clarity.*

To make the motivation clearer we have added the following text (see below in in bold) to the sentence at the beginning of the paragraph that the reviewer mentions:

Advancing from hypothesis testing to parameter estimation, we note that
the UKBB did not recruit families and participants were all adults providing
their own consent¹⁶.

Final Decision Letter:

31st May 2023

Dear Stefania,

I am delighted to say that your manuscript "Studying the genetics of participation using footprints left on the ascertained genotypes" has been accepted for publication in an upcoming issue of Nature Genetics.

Your paper will be published online after we receive your corrections and will appear in print in the next available issue. You can find out your date of online publication by contacting the

Nature Press Office (press@nature.com) after sending your e-proof corrections. Now is the time to inform your Public Relations or Press Office about your paper, as they might be interested in promoting its publication. This will allow them time to prepare an accurate and satisfactory press release. Include your manuscript tracking number (NG-TR59513R2) and the name of the journal, which they will need when they contact our Press Office.

Please note that *Nature Genetics* is a Transformative Journal (TJ). Authors may publish their research with us through the traditional subscription access route or make their paper immediately open access through payment of an article-processing charge (APC). Authors will not be required to make a final decision about access to their article until it has been accepted. [Find out more about Transformative Journals](https://www.springernature.com/gp/open-research/transformative-journals)

Authors may need to take specific actions to achieve <https://www.springernature.com/gp/open-research/funding/policy-compliance-faqs> compliance with funder and institutional open access mandates. If your research is supported by a funder that requires immediate open access (e.g. according to [Plan S principles](https://www.springernature.com/gp/open-research/plan-s-compliance)) then you should select the gold OA route, and we will direct you to the compliant route where possible. For authors selecting the subscription publication route, the journal's standard licensing terms will need to be accepted, including <https://www.nature.com/nature-portfolio/editorial-policies/self-archiving-and-license-to-publish>. Those licensing terms will supersede any other terms that the author or any third party may assert apply to any version of the manuscript.

Please note that Nature Portfolio offers an immediate open access option only for papers that were first submitted after 1 January, 2021.

If you have not already done so, we invite you to upload the step-by-step protocols used in this manuscript to the Protocols Exchange, part of our on-line web resource, natureprotocols.com. If you complete the upload by the time you receive your manuscript proofs, we can insert links in your article that lead directly to the protocol details. Your protocol will be made freely available upon publication of your paper. By participating in natureprotocols.com, you are enabling researchers to more readily reproduce or adapt the methodology you use. [Natureprotocols.com](http://natureprotocols.com) is fully searchable, providing your protocols and paper with increased utility and visibility. Please submit your protocol to <https://protocolexchange.researchsquare.com/>. After entering your nature.com username and password you will need to enter your manuscript number (NG-TR59513R2). Further information can be found at <https://www.nature.com/nature-portfolio/editorial-policies/reporting-standards#protocols>

Sincerely,

Michael Fletcher, PhD
Senior Editor, Nature Genetics

ORCID: 0000-0003-1589-7087